# CoastFLOOD: A High-Resolution Model for the Simulation of Coastal Inundation Due to Storm Surges

**Christos Makris \*, Zisis Mallios, Yannis Androulidakis and Yannis Krestenitis**

Laboratory of Maritime Engineering and Maritime Works, Division of Hydraulics and Environmental Engineering, School of Civil Engineering, Aristotle University of Thessaloniki, 54124 Thessaloniki, Greece; zmallios@civil.auth.gr (Z.M.)

**\*** Correspondence: cmakris@civil.auth.gr

**Abstract:** Storm surges due to severe weather events threaten low-land littoral areas by increasing the risk of seawater inundation of coastal floodplains. In this paper, we present recent developments of a numerical modelling system for coastal inundation induced by sea level elevation due to storm surges enhanced by storm tides. The proposed numerical code (CoastFLOOD) performs high-resolution (5 m × 5 m) raster-based, storage-cell modelling of coastal inundation by Manning-type equations in decoupled 2-D formulation at local-scale (20 km × 20 km) lowland littoral floodplains. It is fed either by outputs of either regional-scale storm surge simulations or satellite altimetry data for the sea level anomaly. The presented case studies refer to model applications at 10 selected coastal sites of the Ionian Sea (east-central Mediterranean Sea). The implemented regular Cartesian grids (up to 5 m) are based on Digital Elevation/Surface Models (DEM/DSM) of the Hellenic Cadastre. New updated features of the model are discussed herein concerning the detailed surveying of terrain roughness and bottom friction, the expansion of Dirichlet boundary conditions for coastal currents (besides sea level), and the enhancement of wet/dry cell techniques for flood front propagation over steep water slopes. Verification of the model is performed by comparisons against satellite ocean color observations (Sentinel-2 images) and estimated flooded areas by the Normalized Difference Water Index (NDWI). The qualitative comparisons are acceptable, i.e., the modelled flooded areas contain all wet area estimations by NDWI. CoastFLOOD results are also compared to a simplified, static level, "bathtub" inundation approach with hydraulic connectivity revealing very good agreement (goodness-of-fit > 0.95). Furthermore, we show that proper treatment of bottom roughness referring to realistic Land Cover datasets provide more realistic estimations of the maximum flood extent timeframe.

**Keywords:** coastal flooding; numerical modelling; storm surge; sea level elevation; inundation maps; Manning coefficient; raster grid

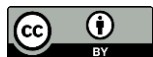

## 1. Introduction

Storm surges, i.e., a (spatially) broad-scale and abnormal elevation of sea level in coastal areas due to severe weather events (storms, tropical cyclones, hurricanes, typhoons, etc.), threaten low-land littoral areas by increasing the risk of seawater inundation of coastal floodplains and low-lying urban environments [1]. This threat intensifies when high seas due to storm surges (meteorological residual of sea level rise) are combined with high astronomical tides (storm tides) [2]. The projected possible Mean Sea Level Rise (MSLR) due to probable future environmental changes in the climatic scale can also further stimulate the intensity of such phenomena on the coastal zone. Moreover, future projections of cyclone characteristics have shown that detrimental extreme events of marine storminess, such as heavy precipitation, windstorms, and storm surges, are strongly associated with each other and can drive coastal flood hazards in a combined way over the Mediterranean basin [3,4]. Thus, storms may affect the sea level elevation on the

shoreline/waterfront in two ways: (a) by increasing the Sea Surface Height (SSH) due to the inverted (or inverse) barometer effect (low-barometric atmospheric pressure), and (b) by winds pushing seawater onshore. The floodwater in coastal areas can overtop physical obstacles or artificial barriers (e.g., dunes and knolls or seawalls, levees, embankments, and armoured slopes), and consequently inundate large parts of inland rural and urban areas. Coastal inundation is mainly responsible for land loss, erosion, damages to onshore infrastructure and properties, environmental degradation of coastal aquatic ecosystems, saltwater intrusion in coastal aquifers, and, occasionally, human casualties, etc.

*1.1. Research Theme*

The most prominent natural hazard induced by episodic bursts of SSH or (semi-)permanent long-term MSLR is coastal flooding and/or the inundation of littoral lowlands, with various significant implications for the coastal communities and environments [5]. Several studies have presented investigations of the coastal vulnerability due to the impact of MSLR and storm surges, related flood hazards, and damage assessment in the eastern and western Mediterranean littorals; e.g., Moroccan and Egyptian coasts [6–8]; NE Mediterranean coastal zone [9]; Ebro river delta in Spain (NW Mediterranean) [10]; coastal inundation risk assessment due to combined land subsidence and MSLR in southern Italy [11]; estimation of 49 cultural World Heritage Sites in low-lying Mediterranean coastal areas until 2100 [12]; potential MSLR-induced inundation in the central Mediterranean (Malta) for susceptibility assessment and risk assessment scenarios to lead policy action [13]. Hauer et al. [14] assessed the exposure of the U.S. population to coastal flooding due to MSLR, while Kulp and Strauss [15] showed that the latest developments in assessments and error corrections of Digital Elevation Models (DEM) have induced a rise in estimates of global vulnerability to MSLR and related coastal flooding. A robust model implementation for such phenomena producing realistic inundation hazard maps is crucial in terms of coastal management, the study of risk, flood hazard mitigation, first-level response to disaster, and decision support.

In this paper, we present recent developments of a numerical model for coastal inundation on littoral floodplains induced by sea level elevation due to storm surges (CoastFLOOD) [16], potentially enhanced by astronomical tides (and MSLR, not investigated in this paper). The model performs numerical simulations of hydraulic flood flow on inland coastal domains covering local-scale areas up to a few hundred km² [17,18]. The inundation model can be forced either with sea level observations (e.g., in situ measurements from tide-gauges and satellite-derived data) or with modelling outputs of regional-scale simulations for storm surges [17]. The High-Resolution Storm Surge (HiReSS) [19] has been used in operational forecast mode for short-term marine weather predictions (sea level and currents) [20,21], providing boundary conditions for CoastFLOOD simulations over adjacent coastal zones [17]. Furthermore, it has been applied as the Mediterranean Climatic Storm Surge (MeCSS) model in climatic studies for long-term hindcasts [9] or future projections of storm surge patterns in the Mediterranean Sea [22–24].

CoastFLOOD performs detailed modelling of the rather shallow and slow process of seawater uprush and flood routing due to episodic, mid- or long-term sea level elevation, i.e., induced by storm surges/tides. It is a very fine resolution, raster-based, 2-D horizontal, mass balance flood model for coastal inlands, following the simplified concept of a reduced complexity form of the Shallow Water Equations (SWEs) running on a storage-cell GIS domain [25–27]. Only the large-scale low-frequency phenomena of coastal inundation due to storm surges and tides are simulated by the model, which does not consider the high-frequency processes of coastal flooding due to wave run-up. The storm-induced SSH on the coastline feeds the seawater surge on the littoral floodplain via a set of 2-D decoupled Manning-type flow equations. The floodwater inundation on the coastal terrain is simulated on a very high resolution ($dx$ = 2–5 m) ortho-regular Cartesian raster grid. Land elevation data are derived by the post-processing of available DEM datasets by the Hellenic Cadastre [28], available in 4600 × 3600 m² ground tiles by the projection of the

Hellenic Geodetic Reference System 1987 (HGRS87). The detailed features of the model are discussed in Section 2.

### 1.2. Literature Review

Numerous 2-D horizontal models exist for the simulation of the mid- to long-term coastal inundation due to storm tides with or without the influence of MSLR. The most representative and established flood inundation model suites have been developed for river flooding and fluvial inundation but can be also used for seawater inundation in coastal areas (Table 1).

**Table 1.** Representative 2-D inundation model suites for river flooding, also used in coastal areas.

| Model | References | Concept and Applications |
|---|---|---|
| LISFLOOD-FP CAESAR-LISFLOOD | [29–34] | Reduced complexity inertial formulation of the SWEs leading to 2-D horizontally decomposed Manning-type volumetric flow equations mainly applied in coastal areas with rivers, optionally coupled to a Landscape Evolution Model (LEM) simulating the geomorphic development of flood basins. |
| MIKE FLOOD 2D | [35,36] | The well-known proprietary flood model suite combining different modules, MIKE URBAN, MIKE 11, and MIKE 21, for urban sewerage systems overflows, river/channel flood discharges, and coastal drivers applied in coastal cities. |
| HEC-RAS 2D | [37,38] | The classic non-commercial flood modelling system combining 1-D/2-D for river flood flow and fluvial plain inundation with the coastal floodplain extent due to sea level changes simulated with the ADCIRC hydrodynamic model. |
| SOBEK-2DFLOW (Overland Flow) | [39,40] | Based on complete Saint Venant equations; a fully hydrodynamic 2-D simulation engine for steep floodwater fronts, wetting and drying processes, subcritical and supercritical flow, including rainfall runoff; applications combine pluvial floods with storm surge influence in urban areas. |
| FLO-2D | [41] | Reduced complexity 2-D Manning-type volumetric flow storage cell simulator coupled to JMA storm surge model. |
| Multi-Scale Nested MSN-Flood model | [42,43] | High-resolution multi-scale modelling of coastal flooding due to tides, storm surges, and river flows specifically for urban coastal inundation. |
| FloodMap-Inertia | [44] | An urban flood inundation model neglecting the convective acceleration term in the momentum equation, coupled to ADCIRC for sea level on its coastal boundary, assuming that the floodplain is filled with water by an embankment-type of river-littoral boundaries essentially acting as a continuous, broad-crested weir, through which flow exchange occurs between channel and floodplain. |
| Floodity | [45] | An anisotropic dynamic mesh optimization (DMO) technique for 2-D double control-volume and a finite element adaptive mesh model for urban coincidental flood modelling. |
| Delft Flooding System (Delft-FLS) | [46] | Overland flow simulation by the 2-D Saint-Venant equations on a rectangular, staggered grid with a finite difference method employing a shock-capturing numerical scheme suitable for rapidly modeling varying flows over rough |

| | | |
|---|---|---|
| | | terrains, including flow through defense breaches and around buildings (minimum depth of 0.01 m distinguishes "dry" from "flooded" cells). |
| Unstructured Tidal, Residual, Intertidal, Mudflat version 2 (UnTRIM²) | [47] | A semi-implicit, Eulerian-Lagrangian finite difference/finite volume model, governed by 3-D SWEs with Boussinesq approximation solved for free surface elevation, water velocities (and salinity) in a Cartesian coordinate system on an unstructured orthogonal grid including both 3-D barotropic and baroclinic processes (tide, wind, and gravitationally-driven circulation). |
| Sea, Lake, and Overland Surges from Hurricanes (SLOSH) | [48,49] | A polar-grid storm surge model with gradually varying cell sizes covering a basin extending from the possibly flooded inland area up to deep water, with a dedicated computation scheme on a B-grid to simulate wetting and drying processes. Water surface elevation differences act as hydraulic load for floodwater propagation to the surrounding grid cells. |
| Stevens Estuarine And Coastal Ocean Model (sECOM) | [50,51] | A successor model to the Princeton Ocean Model (POM) family of models; a 3-D, free surface, hydrostatic, primitive equation estuarine and coastal ocean circulation model with a wetting-drying flood model approach along free moving boundaries. |
| Xie-Pietrafesa-Peng (XPP) model | [52,53] | A HM-C mass-conserving inundation wetting/drying scheme coupled to POM-3D. |
| Cellular Automata (CA) modules | [54,55] | A simplified, grid-based, Saint-Venant equations, 2-D shallow hydrodynamic module discretized in time, space, and state, with local spatial interaction and temporal causality, also optionally running on a triangular finite element mesh. |

Within this modelling framework, Hubbert and McInnes [56] introduced a storm surge inundation model through treatment of the coastal boundary configured to pass through the velocity grid points on the staggered grid in a stepwise manner and define wet/dry cells in inland areas based on a predefined threshold of local water depth on each cell. Nevertheless, many researchers have discussed the practical need for reduced physical complexity approaches [57] to adequately simulate 2-D flood inundation [58] compared to the full-scale 3-D hydrodynamic or 2-D SWE modelling of complex flood flow routing [59–66]. The latter mainly applies to 2-D river floodplain flows, but it is equally valid for 2-D coastal plain flooding either by waves or storm tides. Nevertheless, proper testing and validation of flood inundation models [67] intended for specific hydrodynamic, hydraulic, or hydrological processes dictate the concept of equifinality in model implementation [68]. Our case outlook is to adequately simulate (in terms of robustness and computational resources availability) the coastal inundation extents (including a fine 2-D horizontal local distribution of water heights) and the response times of coastal flood maxima within an oversimplified methodological framework minimizing the uncertainty of parametric analysis and dependence on unreliable or insufficient (topographic and land use) input [69].

### 1.3. Research Incentive

The proposed model follows the conceptual framework of reduced complexity flood inundation approaches on high-resolution computational grids in a way to balance between the reliability and practicality of applications in the coastal zone [41,44,70–72]. Hence, we introduce a recently developed in-house model (CoastFLOOD) specifically designed for fine-scale hydraulic flooding of seawater in littoral areas. It is specifically built to work in operational mode, meeting the need to be easily coupled to a coarser large-sale storm surge model (e.g., HiReSS) written in the same programming language and using

similar coding modules and job execution tactics. Our goal was to further formulate proper and detailed input for spatially varying Manning roughness coefficients, especially fitted to 2-D coastal floodplains. This way we can uphold the physical properness, assist the calibration process and the robust performance of the model in a timely manner for operational forecasting, and engineer consulting purposes [73,74].

The scope of the study is to further evaluate the impact of detected sea level variations (either by modelling or monitoring procedures) on seawater inundation patterns over several characteristic regions of the Greek coastal zone. Kulp and Strauss [75] have discussed the necessity to minimize errors in DEMs to avoid underestimations of coastal vulnerability due to MSLR-induced flooding. Therefore, the CoastFLOOD model is tested in tandem with an updated dataset of land elevation derived from a DEM with a resolution of $dx$ = 2–5 m that covers 10 selected lowland areas along the Ionian Sea coastline. These have been identified as highly impacted areas by intense flooding events in the past [17].

The model domains include various urban and suburban settlements, rural coastal plains, environmentally protected areas (lagoons, estuaries, wetlands, and aquatic habitats), touristic infrastructure areas, recreational coastal zones with sandy beaches, and coastal regions accumulating several activities (e.g., aquaculture, fisheries, navigational transportation, seaport commerce, etc.). Coastal inundation hazard maps are produced to estimate the littoral flooding variability over the Greek coastline. Model validation is performed for the operational forecast mode of CoastFLOOD simulations against fine-scale satellite observations (by Sentinel-2 images at 10-m resolution), producing the Normalized Difference Water Index (NDWI) [76,77]. Model results are also compared to a static level, enhanced "bathtub" inundation approach with "eight-side rule" hydraulic connectivity [78–82].

New model features are also presented, concerning the detailed surveying of terrain roughness and bottom friction, the expansion of Dirichlet boundary conditions for coastal currents (besides sea level), the enhancement of a wet/dry cell technique for flood front propagation over positive/negative steep terrain slopes, etc.

All of the methodological information regarding the model setup, parameterization features, numerical schemes, and computational grids are thoroughly described in Section 2. Case study characteristics and datasets for model validation are presented in Section 3. The results regarding coastal flooding are analysed in Section 4. A discussion of the study findings is presented in Section 5, followed by a section of concluding remarks (Section 6).

## 2. Methodology

### 2.1. Conceptual Approach of Storm Surge Inundation

The basic concept of our modelling approach refers to implementing a set of simplified continuity and momentum conservation equations for the simulation of rather shallow and slow inundation processes [16,18]. These are primarily (or solely) driven by the sea level elevation on the coastline and secondarily by the estimation of the barotropic coastal current as long as it has an onshore direction. Therefore, we simulate the sluggish seawater flooding on the low-lying coastal areas that is induced by a slow surface flow due to storm surge, unlike the fast-evolving undulating flows that are caused by swell and wind-wave action on the coast.

The model's advantageous feature is that it can be applied at very high spatial resolutions (e.g., $dx$ = 1–5 m) for a geophysical-scale flow, while the feeding input of $SSH$, acting as the hydraulic head that defines the piezometric load on the boundary conditions, can be of wider scales (e.g., O($Dx$) = 1–10 km) [17]. This allows for a practically efficient compromise between the validity of representation of the governing physics and operational model adequacy for hydraulic engineering problems in large-scale environmental flows. The chosen raster modelling approach adopts a (horizontally) decomposed uniform flow approximation for coastal floodplain flow, which is mainly dominated by gravity and friction to calculate the momentum balance [18]. This is a reasonable

approximation for gradually evolving (laminar) flows over mild sloping floodplains in rural or natural areas; however, it may be an oversimplification for unsteady hydraulic flows in complex urban environments, where turbulent effects play a starker role in rapidly varying topographies. Neglecting pressure and/or inertial terms of the momentum equation may lead to erroneous representation of the floodwater flow characteristics in the built environment. Nevertheless, the assumed model approach has been shown in the past to be able to adequately predict the horizontal extents of inundation and the floodwater height in inland areas even if they lie in urban regions. The simplified kinematic scheme of the Manning-type hydraulic flow allows for numerical applications on regular gridded domains of large areas, typically incorporating up to $15 \times 10^6$ model grid cells, testing the limits of modern available computational resources.

### 2.2. Numerical Model for Hydraulic Flow in Coastal Flooding

CoastFLOOD [16–18] is an in-house numerical model built on a FORTRAN-95 code, that solves the depth-averaged, 2-D horizontal, mass balance, flood flow equations [25–27,29]. These have produced a series of 2-D floodplain applications [70,72,83] particularly implemented in coastal case studies [84–88]. The latest version of the model, presented herein, has been enhanced in terms of bottom roughness treatment to include cases in:

(a) Rural plains with agricultural zones and farmlands, wild flora or natural vegetated fields, forests, bare or stony lands, pastures, and grasslands, etc.;

(b) Wet inland areas, such as shores, estuaries, lagoons, river deltas, beaches, etc.;

(c) Urban and sub-urban areas with engineered coasts, built waterfronts, ports and coastal protection structures, roads, highways, railway networks, dense building constructions or open spaces and parks, mildly or highly developed built environments, etc.

The robustness of similar model approaches (e.g., LISFLOOD-FP, FLO-2D, Floodmap) has been validated and applied in 2-D floodplains in coastal areas or fluvial landforms [29,80,83,89]. CoastFLOOD also follows a simplistic finite difference scheme for hydraulic flow inundation, running on very fine resolution raster grids, able to reproduce the surge-induced 2-D flood on the coast [17,18]. Propagation of the floodwater front is decomposed in two horizontal Cartesian x- and y-directions, allowing for discrete zonal and meridional components of the flow, respectively, for inland flood routing [26,71,90].

The simplified form of the 2-D equations for conservation of mass (continuity) and momentum are discretized over an ortho-regular grid of rectangular cells (Figure 1a), in order to reproduce the evolution of a 2-D Manning-type flow between neighbouring cells over the entire floodplain [58]. The floodwater flow between adjacent cells is mainly driven by the hydraulic head created by the inter-cell difference of water surface height in all four cardinal directions of the horizon (Figure 1). Thus, the continuity equation relates the floodwater volume of an arbitrary cell to the volumetric flows in and out of it, during a typical timestep of the numerical solution. This is written in the form of generic volumetric (Equation (1)), (analytic) spatially discretized volumetric and piezometric head (Equations (2) and (3)), grid- and time-discretized (Equation (4)) equations, as:

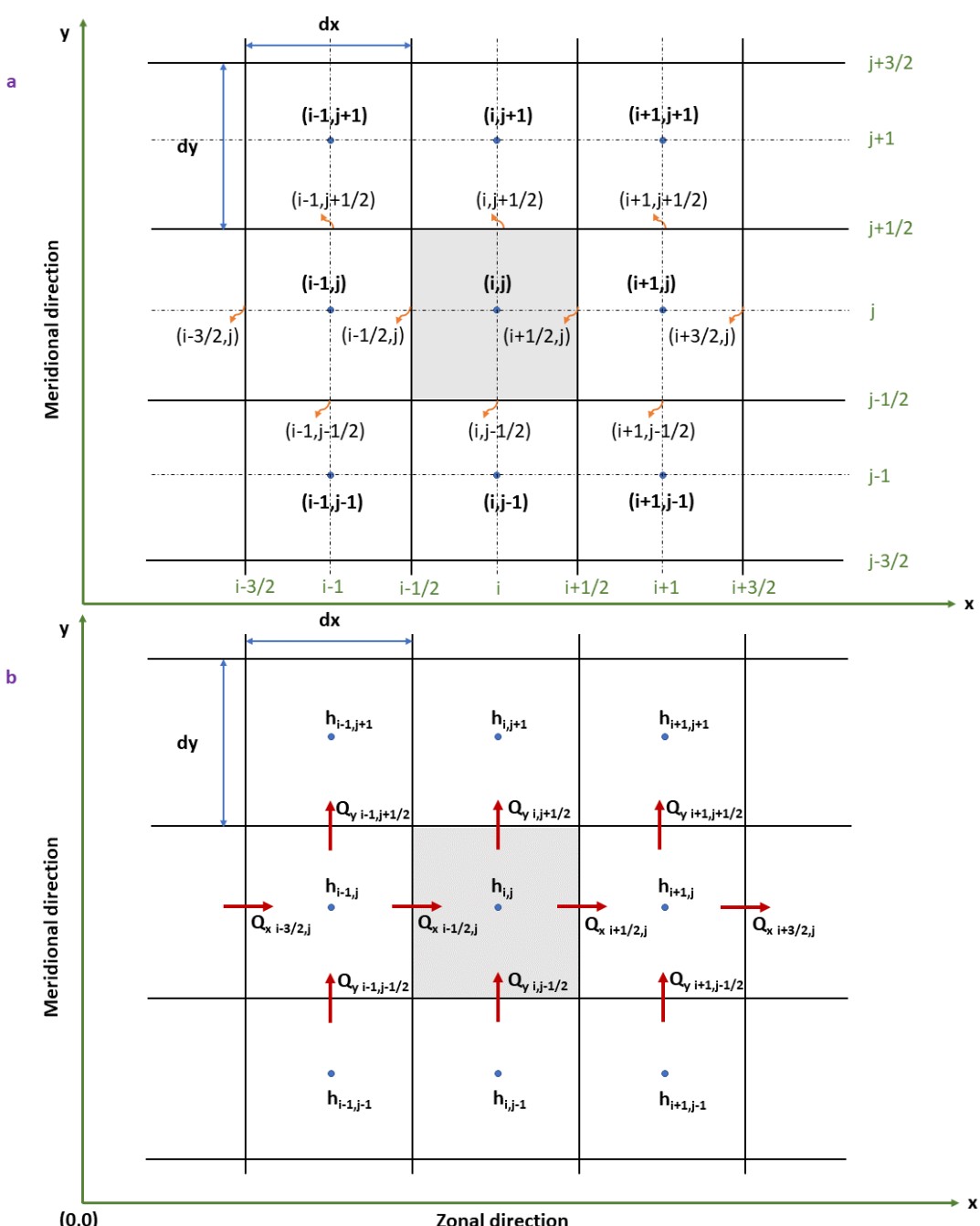

**Figure 1.** Depiction of the prototype Cartesian raster grid formulating a typical computational do-
main in CoastFLOOD model; (**a**) discretization of the staggered grid cells (at their centres and faces)
with *dx* and *dy* dimensions, over an i−j coordinate system on the Cartesian *x*- and *y*-directions (zonal
and meridional directions of the horizon); (**b**) notation of scalar parameter floodwater height h at
the centres of the grid cells and decoupled vectorial parameter volumetric flow rate, $Q_x$ and $Q_y$,
between adjacent cells (at their interfaces). The shaded cell is the main cell of parametric numerical
calculation at each timestep. Arrow directions represent the positive values of flow pathways be-
tween grid cells; i.e., from floodwater flow upstream areas to downstream ones.

$$\frac{\partial V}{\partial t} = Q_x^{in} - Q_x^{out} + Q_y^{in} - Q_y^{out} \tag{1}$$

$$\frac{\partial V_{i,j}}{\partial t} = Q_{x_{i-1/2,j}} - Q_{x_{i+1/2,j}} + Q_{y_{i,j-1/2}} - Q_{y_{i,j+1/2}} \tag{2}$$

$$\frac{\partial h_{i,j}}{\partial t} = \frac{Q_{x_{i-1/2,j}} - Q_{x_{i+1/2,j}} + Q_{y_{i,j-1/2}} - Q_{y_{i,j+1/2}}}{\partial x \cdot \partial y} \tag{3}$$

$$h_{i,j}^{t'} = h_{i,j}^{t} + dt \cdot \frac{Q_{x_{i-1/2,j}}^{t} - Q_{x_{i+1/2,j}}^{t} + Q_{y_{i,j-1/2}}^{t} - Q_{y_{i,j+1/2}}^{t}}{dx \cdot dy} \quad \text{or}$$

$$h_{i,j}^{t'} = h_{i,j}^{t} + dt \cdot \left( \left( \theta \cdot \frac{Q_{x_{i-1/2,j}}^{t} - Q_{x_{i+1/2,j}}^{t} + Q_{y_{i,j-1/2}}^{t} - Q_{y_{i,j+1/2}}^{t}}{dx \cdot dy} \right) + \left( (1-\theta) \cdot \frac{Q_{x\,i-1/2,j}^{t'} - Q_{x\,i+1/2,j}^{t'} + Q_{y\,i,j-1/2}^{t'} - Q_{y\,i,j+1/2}^{t'}}{dx \cdot dy} \right) \right) \tag{4}$$

where, $V$ is the volume with $V_{ij}$ referring to cell $(i,j)$, $i$ and $j$ being the x- and y-directions of the Cartesian grid; +1/2 in indexing denotes the intercell positioning of flow parameters; $t$ is the time and $dt$ the timestep of temporal discretization (hence, $t' = t + dt$ at the following timestep in the solution scheme); $Q_x$ and $Q_y$ are the volumetric flow rates between adjacent floodplain cells in the zonal $x$- and meridional $y$-directions of the Cartesian grid, respectively; $Q^{in}$ and $Q^{out}$ are the incoming and outgoing volumetric flow rates in a typical grid cell within the generic representation of the equations; $h$ is the local floodwater height above each grid cell's land elevation, $z$; $dx$ and $dy$ are the cell dimensions in the zonal $x$- and meridional $y$-directions of the Cartesian grid, respectively; $\theta$ is a numerical weighting coefficient, which determines whether the equations are fully solved or partially implicitly for $\theta < 1$ or explicitly for $\theta = 1$ [58]. The explicit scheme is the norm, but both options are provided in the CoastFLOOD model. Note that the scalar magnitude of local water height, $h$, is calculated on each cell's centre or any adjacent cell's centre, e.g., $h_{i,j}$ or $h_{i+1,j}$ or $h_{i,j-1}$, while the vectorial magnitude of flow rate, $Q$, is calculated on either of the side faces of each cell or either of the side faces of any adjacent cell, e.g., $Q_{i-1/2,j}$ or $Q_{i,j+1/2}$; hence, practically rendering the solution scheme on a staggered grid (Figure 1b).

This way, we allow for each floodplain grid element to function as an individual storage cell, letting a simplified formulation of the momentum equation derive inter-cell fluxes. Equations in x- and y-directions can be written in the form of an analytic kinematic function based on Manning's law, permitting the decomposed calculation of the flow rate in each grid cell, reading in generic form:

$$Q = \frac{h_{flow}^{5/3}}{n} \cdot \left( \frac{h_{upstream} - h_{downstream}}{CellWidth_{zonal}} \right)^{1/2} \cdot CellWidth_{meridional} \tag{5}$$

where, $CellWidth_{zonal} \equiv dx$ and $CellWidth_{meridional} \equiv dy$ are generic notations of cell dimensions in horizontal directions; indices $upstream$ and $downstream$ refer to generic representations of, e.g., $(i-1,j)$ and $(i+1,j)$ cells for $(i,j)$ central element of numerical calculation at each timestep; $n$ is the Manning's coefficient of roughness for bed friction inclusion; $h_{flow}$ is the flow depth between two adjacent cells, i.e., defined as the difference of the highest floodwater surface elevation from Mean Sea Level (MSL), $H$, minus the maximum bed elevation, $z$, between two neighbouring cells (Figure 2).

The spatially discretized version of Equation (5) further reads:

$$Q_x^{in} = \frac{h_{flow_{x,in}}^{5/3}}{n} \cdot \left( \frac{h_{i-1,j} - h_{i,j}}{dx} \right)^{1/2} \cdot dy \,, Q_x^{out} = \frac{h_{flow_{x,out}}^{5/3}}{n} \cdot \left( \frac{h_{i,j} - h_{i+1,j}}{dx} \right)^{1/2} \cdot dy$$

$$Q_y^{in} = \frac{h_{flow_{y,in}}^{5/3}}{n} \cdot \left( \frac{h_{i,j-1} - h_{i,j}}{dy} \right)^{1/2} \cdot dx \,, Q_y^{out} = \frac{h_{flow_{y,out}}^{5/3}}{n} \cdot \left( \frac{h_{i,j} - h_{i,j+1}}{dy} \right)^{1/2} \cdot dx \tag{6}$$

where, again, indices $in$ and $out$ denote incoming and outgoing flows.

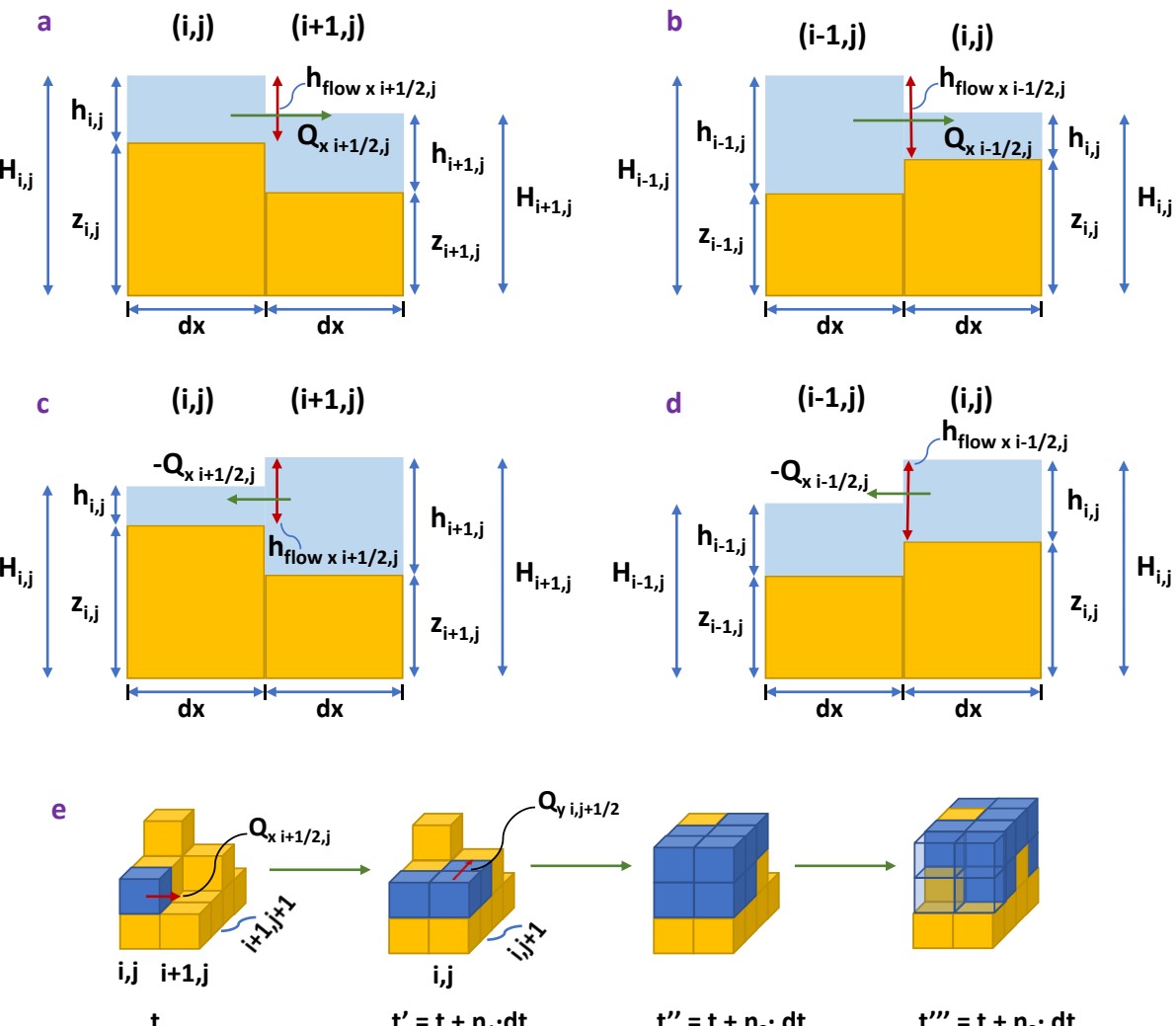

**Figure 2.** Depiction of flood front propagation over typical grid cells in the CoastFLOOD model's 2-D x-z plane (graphs (**a**,**b**)) and wet/dry cell expansion in pseudo-3-D projection (graph (**c**)). (**a**–**d**) Schematic representation of $Q_x$ and $h_{flow}$, i.e., the flow depth between two adjacent cells, defined as the difference of the highest floodwater surface elevation from MSL zero-level, H, minus the maximum bed elevation, z, between two neighbouring cells either (*i*,*j*) and (*i*+1,*j*) in graphs a and c, or (*i*−1,*j*) and (*i*,*j*) in graphs (**b**,**d**). (**e**) Illustrative representation of progressive inundation front by discretized floodwater flow propagation and encroachment on an elevating model grid with explicitly modelled micro-topography at arbitrary ($n_1$–$n_3 \cdot$ t) timesteps. Yellow-brown cube-cells refer to ground, while blue ones refer to floodwater.

The spatiotemporally discretized form of Equation (6) corresponding to placement on a typical model grid (Figure 1b) is written as:

$$Q_{x_{i-1/2,j}}^t = \frac{h_{flow\,x_{i-1/2,j}}^{t\,5/3}}{n} \cdot \left(\frac{h_{i-1,j}^t - h_{i,j}^t}{dx}\right)^{1/2} \cdot dy, \quad Q_{x_{i+1/2,j}}^t = \frac{h_{flow\,x_{i+1/2,j}}^{t\,5/3}}{n} \cdot \left(\frac{h_{i,j}^t - h_{i+1,j}^t}{dx}\right)^{1/2} \cdot dy$$

$$Q_{y_{i,j-1/2}}^t = \frac{h_{flow\,y_{i,j-1/2}}^{t\,5/3}}{n} \cdot \left(\frac{h_{i,j-1}^t - h_{i,j}^t}{dy}\right)^{1/2} \cdot dx, \quad Q_{y_{i,j+1/2}}^t = \frac{h_{flow\,y_{i,j+1/2}}^{t\,5/3}}{n} \cdot \left(\frac{h_{i,j}^t - h_{i,j+1}^t}{dy}\right)^{1/2} \cdot dx$$

(6)

where *t* is the current time and can be also substituted by *t*′ to represent the needed values in Equation (4), and $h_{flow}$ in absolute discretised notation (Figure 2) can be calculated based on the equation:

$$h_{flow_{x_{i-1/2,j}}} = \left(max\{H_{i-1,j}, H_{i,j}\} - max\{z_{i-1,j}, z_{i,j}\}\right) \tag{7}$$

The exponent $h_{flow}5/3$ refers to a Manning law approach for the flood propagation and can be used under the assumption of a uniform laminar flow over a flat rectangular cell ($dx = dy$ wide grid element) of constant depth.

Equations (5)–(7) describe the reduced complexity versions of the momentum equations, which are typically based on a semi-analytical approach for hydraulic flows, such as the aforementioned Manning-type equation. Alternately, the user can choose to incorporate the 2-D finite difference approximation of a similar equation for diffusive waves [58]:

$$Q^t_{x_{i-1/2,j}} = \frac{\frac{h^{t\,5/3}_{flow_{x_{i-1/2,j}}}}{n}\left(\frac{h^t_{i-1,j}-h^t_{i,j}}{dx}\right)^{1/2}\cdot dy}{\left(\left(\frac{h^t_{i-1,j}-h^t_{i,j}}{dx}\right)^{1/2}+\left(\frac{h^t_{i,j-1}-h^t_{i,j+1}}{dxdy}\right)^{1/2}\right)^{1/4}}, Q^t_{x_{i+1/2,j}} = \frac{\frac{h^{t\,5/3}_{flow_{x_{i+1/2,j}}}}{n}\left(\frac{h^t_{i,j}-h^t_{i+1,j}}{dx}\right)^{1/2}\cdot dy}{\left(\left(\frac{h^t_{i,j}-h^t_{i+1,j}}{dx}\right)^{1/2}+\left(\frac{h^t_{i,j-1}-h^t_{i,j+1}}{dxdy}\right)^{1/2}\right)^{1/4}} \tag{8}$$

$$Q^t_{y_{i,j-1/2}} = \frac{\frac{h^{t\,5/3}_{flow_{y_{i,j-1/2}}}}{n}\left(\frac{h^t_{i,j-1}-h^t_{i,j}}{dy}\right)^{1/2}\cdot dx}{\left(\left(\frac{h^t_{i,j-1}-h^t_{i,j}}{dy}\right)^{1/2}+\left(\frac{h^t_{i-1,j}-h^t_{i+1,j}}{dxdy}\right)^{1/2}\right)^{1/4}}, Q^t_{y_{i,j+1/2}} = \frac{\frac{h^{t\,5/3}_{flow_{y_{i,j+1/2}}}}{n}\left(\frac{h^t_{i,j}-h^t_{i,j+1}}{dx}\right)^{1/2}\cdot dy}{\left(\left(\frac{h^t_{i,j}-h^t_{i,j+1}}{dx}\right)^{1/2}+\left(\frac{h^t_{i-1,j}-h^t_{i+1,j}}{dxdy}\right)^{1/2}\right)^{1/4}} \tag{9}$$

*2.3. Time Discretization—Numerical Schemes*

The abovementioned discretized Equations (4) and (7)–(10) are solved with the use of appropriate boundary and initial conditions using certified numerical techniques. CoastFLOOD incorporates (user-identified) solvers that implement either an explicit ($\theta$ = 1) forward-time and centered-space (FTCS) finite difference scheme or an implicit ($\theta < 1$) backward-time and centered-space (BTCS) algorithm to obtain predictions of $Q_x$, $Q_y$, and $h$ at any given timestep. The choice of $\theta$ is a prerequisite from the CoastFLOOD user, resulting in different levels of solution complexity/stability and higher model runtimes for the implicit scheme. For $\theta$ = 1, the $Q$ and $h$, at $t'$ can be explicitly computed by the known quantities at $t$ (floodplain flows $Q$ can be initially calculated by Equations (8)–(11)). Consequently, floodwater depths $h$ can be updated by Equation (4a). Explicit algorithms are preferred for their coding simplicity and straightforward integration schemes on a staggered ortho-regular raster grid. Nevertheless, numerical stability is ensured by very small model timesteps, e.g., $dt < 10$ s, according to the Courant-Friedrichs-Lewy (CFL) criterion, $C$:

$$C = u_x dt/dx < 1 \tag{10}$$

e.g., for $C \le 0.5$, the timestep should practically be $dt \le (0.5h_{i,j}dx^2)/Q_x$, where $u_x = Q_x/A_x$ and $A_x = dx \cdot h_{i,j}$ in a typical grid cell. To ensure numerical stability, the following CFL condition, with $\alpha$ = 0.3–0.7, is proposed by [27,33]:

$$dt_{max} = a \cdot dx / \sqrt{gh_{ij}} < 1 \tag{11}$$

Practically, based on Equation (13), for values of, e.g., $h$ = 0.001–1.5 m and $dx$ = 5 m, the minimum achieved timestep should roughly range between $dt_{max} \approx 35$–0.35 s, respectively (for corresponding $\alpha$ = 0.7–0.3). Nonetheless, the aforementioned $dt$ values refer to an upper threshold value, while even lower timesteps may be needed in the course of cell-by-cell numerical solution. Previous studies have proposed the following adaptive timestep [71], based on the Von Neumann condition, especially for the diffusive wave case, as shown in Equations (9) and (10):

$$dt \leq \frac{dx^2}{4(1-\theta)} => dt = \frac{dx^2}{4} min\left(\frac{2n}{h_{flow_x}^{5/3}}\left|\frac{\partial h}{\partial x}\right|^{1/2}, \frac{2n}{h_{flow_y}^{5/3}}\left|\frac{\partial h}{\partial y}\right|^{1/2}\right) \tag{12}$$

This is supposed to eliminate "chequerboard" numerical oscillations, induced when $dt$ becomes large, which essentially occurs for very low $h_{i,j}$ values and consequently low flood flow rates (and floodwater velocities). However, in the CoastFLOOD model, the practical lower/upper cut-off $dt$ values are set to 0.5 s ≤ $dt$ ≤ 5 min (e.g., for $dx$ = 5 m), allowing for reasonable computational times and the avoidance of lagging in the numerical solution, respectively. Likewise, to avoid further instabilities in the advancing iterations of the numerical solution (notably in high floodwater depths, $h_{i,j}$, or highly uneven elevation levels of adjacent cells), we adopt a flow rate limiter, especially for the most classic case of 2-D floodplain flow being controlled by momentum Equation (7). The flow limiter (minimum $Q$ threshold) can also prevent instabilities in adjacent areas of very large differences in floodwater depth [25]:

$$Q_{x_{i-1/2,j}} = min\left\{calculated\ Q_{x_{i-1/2,j}}, \frac{dxdy\left(h_{i,j}^t - h_{i-1,j}^t\right)}{8dt}\right\}$$

$$Q_{x_{i+1/2,j}} = min\left\{calculated\ Q_{x_{i+1/2,j}}, \frac{dxdy\left(h_{i+1,j}^t - h_{i,j}^t\right)}{8dt}\right\} \tag{13}$$

for a concomitant min/max limiter of floodwater velocity that reads 0.01 m/s ≤ $u_x = Q_x/A_x$ ≤ 5 m/s. Similar equations apply to the y-direction of the flow. With this numerical treatment, the user can actually prevent over- or under-shooting of the numerical solution. The flow limiter essentially ensures that floodwater depth change in an arbitrary cell at $t$ is not adequately large to reverse the y flow entering or exiting the cell at $t'$ [71]. $Q$ values derived by Manning's equation are replaced, when overestimated, with values strictly determined by model domain parameters ($dx$ and $dt$). If a small $dt$ or large $dx$ is chosen, the limiter is nearly eliminated. Therefore, the results of the CoastFLOOD model, like many other storage-cell codes for flood flows, are far from invariant with respect to $dx$ and $dt$. Their optimal choice is a matter of experience, taking into account the extents of the entire case study domain and its low-lying areas, etc. Moreover, this approach may undermine the simulations in terms of correctly predicting the advance of flood fronts and the volume of floodwater in inundated areas [91]. The choice of smaller CFL numbers, $C \ll 0.5$; hence, smaller $dt$ can address this discrepancy.

The coastal flooding phenomena, induced by storm surges, may last from several hours up to a few days, i.e., resulting in simulations of $2-4 \times 10^4$ to $2-5 \times 10^5$ timesteps, for a few hours up to 3 days duration of the studied flood event, given that $dt \leq 1$ s. Depending on the number of inland grid cells to be flooded (e.g., up to $40 \times 10^6$ elements), this means that the estimated computational times range from one hour up to more than half a day on a PC with a 10th generation 12-core Intel® i7-CPU, 10750H, @2.60 GHz, with 64 GB RAM and 1 TB SSD hard disk 860 QVO. For the case of the implicit scheme, where $Q$ and $h$ variables depend on unknown quantities at $t'$, an iterative solution technique (e.g., finite difference Preissmann scheme [92]) adds even more computational burden and time. Of course, the implicit scheme allows for larger timesteps in the O(5–10) mins, given the slow evolution of flood events over inundated plains.

The meridional- and zonal-direction decompositions of the flood flow components allow the derived 1-D flow equations for overland seawater propagation to be numerically and separately calculated for each grid cell face on a typical 2-D raster [90]. This makes the calculation of flood routing an easy task, through the use of a simplistic nearest neighbour or quad-tree search algorithm for the downstream cells. The latter are defined as dry or wet (for $h_{ij} > 0.005$ m) and then they are saved and/or updated in a storage cell matrix at every simulation timestep. To this end, the effective water flow depth between two neighbouring cells, $h_{flow}$, which is defined by the difference between the highest possible

water level in adjacent grid elements and their largest land elevation, $z$ (Figure 2), is not allowed to exceed the maximum threshold of $h_{flow} \leq \max(h_{i,j}) = SSH - z_{i,j}$. The $x$- and $y$-direction decoupling of flood flow propagation may not represent the diffusive nature of the inundation wave spreading on the floodplain; however, it has been shown [83] that more complicated treatments of floodplain flows have yielded no significant improvements compared to reduced complexity models [70] when evaluated against Synthetic Aperture Radars (SAR) data.

*2.4. Computational Domain and Raster Grid*

The numerical grid formulation (terrain discretization) for typical, reduced complexity models of coastal inundation by storm surges follows the trends in the development of high-resolution topographic gridded data. Namely, DEMs represent bare earth or ground surface topography, excluding trees, buildings, and any other surface objects, while Digital Surface Models (DSMs) capture the land surface, including vegetation and manmade structures, such as buildings and infrastructures. DEMs are used to construct the entire model domain (mainly focused on natural areas, rural environments, wild lands, etc.), whilst DSMs are implemented within urban and suburban areas to include the flow obstruction by the built environment.

To firstly identify the low-lying areas along the Greek coastal zone and secondly create the detailed topographical input for the storm tide inundation simulations with CoastFLOOD, the GIS datasets of land elevation were retrieved from the official Greek service for the comprehensive recording of real-estate and property metes-and-bounds [28]. There are two available high-resolution DEMs in coastal and inland regions with spatial resolution $dx = 2$ m and 5 m. The rectangular model domains were produced by post-processing of the available polar coordinate geospatial data in the World Geodetic System 1984 (WGS84) to HGRS87. The DEM's geometric accuracy is less than 0.70 m, while its absolute accuracy is less than 1.37 m with a 95% confidence level [17]. Similarly, the DSM's accuracy is less than 0.32 m, while its absolute accuracy is less than 1 m with a 95% confidence level. The DSM has an even finer resolution of $dx = 0.8$ m, and thus its datasets were extrapolated to fit the fixed model's computational domains of $dx = 2$–5 m.

To avoid the underestimation of the storm surge effect driving the flood flow from any possible convex or crooked part of the coastline (no matter how complex it might be or what orientation the shoreline has in the domain), a cross-type scan of the model grid (N→S and S→N in the meridional direction; W→E and E→W in the zonal direction) is applied in every timestep (Figure 3). This way, the volumetric flow rates' signs (Figure 2c,d) are corrected, based on the propagation of the flood front from all directions of the horizon, and thus the wet/dry storage cell matrix is updated with every possible change in water level of each grid element in the model domain (Figure 3). This is a step forward from traditional coastal inundation modeling that considers flood propagation from only one boundary at a time.

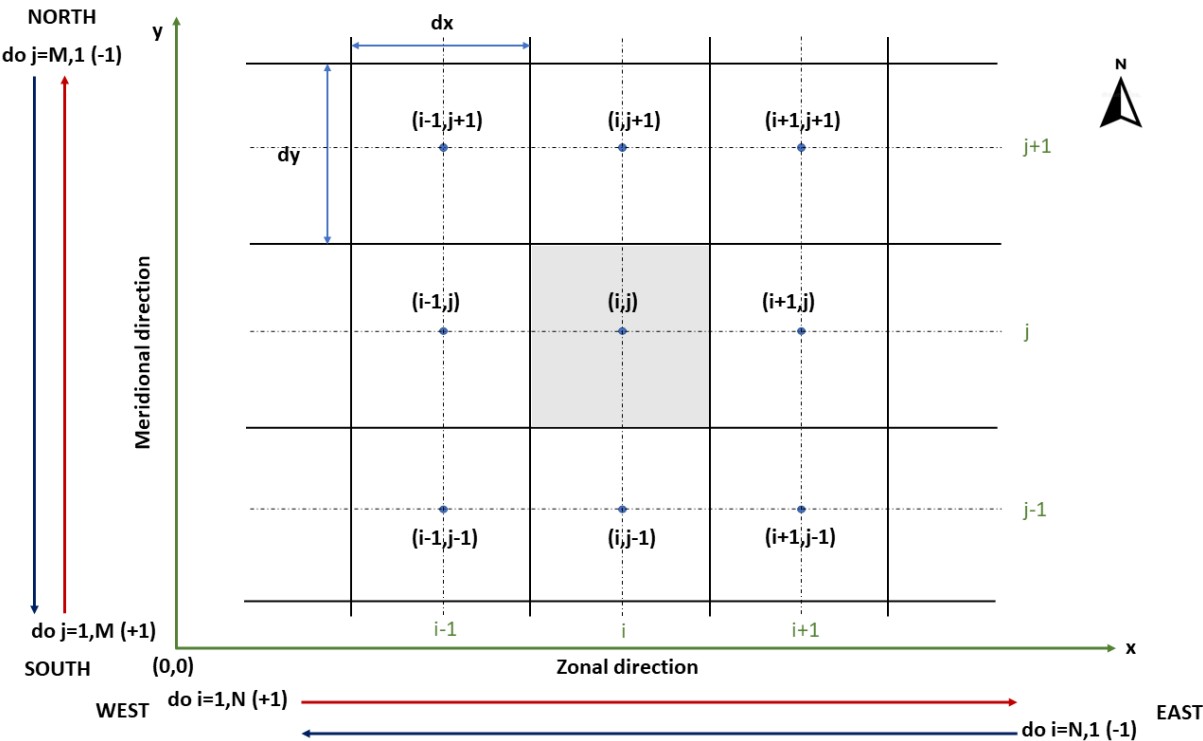

**Figure 3.** Depiction of the cross-type scanning process of the numerical grid by the computational domain in the CoastFLOOD model. Red and blue arrows represent the numerical propagation scan direction of the grid cells on zonal and meridional, *x*- and *y*-axis, for *i* = 1,N and *j* = 1,M (and reverse), respectively, applied at each timestep.

The discrepancies of the DEM/DSM are crucial factors of accuracy in CoastFLOOD simulations of flooded areas, even if the highest available resolution raster grid is used to include topographic details of the natural and urban parts of the coastal domain [93]. CoastFLOOD does not consider the effects of porous bed percolation and ground infiltration, flows in sewerage and drainage systems (e.g., conduits, bridge culverts, wells, shafts, etc.). However, this is not considered a crucial issue, as these constructions are usually saturated with fresh water or drainage/sewage waters from surface runoffs. Coastal inundation usually occurs within a compound flooding incident; i.e., concurrently to river flooding and/or urban flooding due to heavy rainfall and strong runoffs relevant to the storm event also driving the onshore sea surge [37].

### 2.5. Model Parameterization

Bottom friction is the main parameterization feature of reduced complexity flood inundation models. The calculation of hydraulic flows requires the specification of flow resistance or bed roughness in a parametric approach. As the typical model cell's dimensions and depth are assumed to be uniform for each grid element, an effective Manning's bottom roughness coefficient, *n*, at grid unit scale can be determined as a calibration parameter. Seenath [94] thoroughly discussed issues of achieved improvement in prediction modelling of coastal flooding (more in terms of inundated area extents) based on the fine representation of spatially distributed friction over the case study domain against a uniform *n* value all over the model grid.

The CoastFLOOD model incorporates both solutions, i.e., considering the friction effect of the floodplain terrain on the inundation flow either by defining a distributed, effective, grid-scale Manning's *n* on each cell of the model's raster domain or by proposing a representative "global" effective grid-scale *n* coefficient (on the entire domain or large homogenous parts of it). By integrating the relevant literature [74,80,94–101], we created a detailed collective ensemble of proposed Manning coefficient *n* values discretized at 36

increments (Table 2). These values are specifically fitted to 2-D coastal floodplain flows and refer to the most common and less likely types of (natural or artificial) ground material.

Beven [102] argued that a predetermination of bottom roughness parameters at each computational grid point was rarely possible due to scaling problems, i.e., differences between the in situ observation scale and the model grid scale, and other data availability constraints. However, the recent development of the CORINE Land Cover (CLC) inventory [103] provides a robust record of land cover in 44 classes for Europe. CLC uses a minimum mapping unit of 25 ha for areal phenomena and a minimum width of 100 m for linear phenomena; here, we use the latter. CLC is mainly produced on a country/state-level by visual interpretation of fine-resolution satellite imagery from Sentinel-2 and Landsat-8 (for gap filling) products, with the latest time consistency referring to 2017–2018.

Table 3 presents a detailed matching catalogue that we have created for all 36 discrete cases of CoastFLOOD's Manning coefficient listings in Table 2 to the CLC-2018 codes that refer to data of as many possible natural and manmade land cover types. CLC is available in both raster and vector formats; in our case studies, we used the second one, because it is easier to align the land cover data to the constructed model domains. Specifically, for each of the study areas, CLC data were retrieved in QGIS using its boundaries as a reference. Then, a Manning coefficient *n* was assigned to each vector polygon representing a specific land use, using the matching between *n* and land use from Table 2. Finally, a raster image with the same dimensions and spatial resolution as the Manning *n* matrix and the model grid was created. If no CLC are available, a parametric calibration of bottom roughness can be undertaken in order to identify empirical values for the Manning coefficient. Terrain heterogeneities on the sub-grid level can cause discrepancies in the representation of land cover texture, thus Manning's *n* is commonly used as a determinative calibration parameter rather than a physical factor of actual field friction.

**Table 2.** CoastFLOOD 2-D modified floodplain Manning coefficient list.

| A/A | n | Description of Areas' Characteristics |
|---|---|---|
| 1 | 0.001 | open water |
| 2 | 0.0115 | concrete surfaces |
| 3 | 0.010 | rural driveways (dirt road and granules) |
| 4 | 0.012 | urban land uses (asphalt mixtures and other urban surface features: artificial stones, paving blocks, lightweight aggregate concrete), concrete rooftop, playground, yard, barren land |
| 5 | 0.013 | main asphalt roads (national, regional highway networks, autobahns, etc.) |
| 6 | 0.015 | brick terrain, unidentified high and low development urban environment, inland open waters (reservoirs, lakes, ponds, lagoons, estuaries) |
| 7 | 0.017 | city streets (asphalt, concrete, etc.) |
| 8 | 0.018 | unidentified/unclassified urban terrain |
| 9 | 0.020 | clean to gravelly earth pathways (pebbles with a small portion of cobbles), muddy/sandy open waters and sandy terrains, sea bottom (saturated wet sand or silt-sand) and channel beds |
| 10 | 0.030 | bare unidentified/unclassified soil |
| 11 | 0.022 | bare land, stone paved road and ceramic sett, or paving sett pathways |
| 12 | 0.029 | stony cobble lands, pastures, and farmlands |
| 13 | 0.025 | manmade structures, gravel beds and pathways (pebbles with nominal diameter: $d_{n50}$ = 4–64 mm, cobbles: $d_{n50}$ = 64–256 mm) |
| 14 | 0.0375 | cultivated fields and pasture, grassland (including prairies, steppes, plains) |
| 15 | 0.0425 | isolated sand/gravel(mixed) pits, estuary channels, and uneven urban areas |
| 16 | 0.029 | emerged sloping sandy beaches, sand dunes |
| 17 | 0.030 | managed grasslands |

| 18 | 0.0115 | unclassified/unidentified rural areas |
|---|---|---|
| 19 | 0.033 | grass surfaces |
| 20 | 0.035 | short stiff grass areas |
| 21 | 0.0575 | weeds with or without structure |
| 22 | 0.0555 | heavy brush floodplains |
| 23 | 0.040 | arable land plains, heavy/coarse gravel (boulders: $d_{n50}$ >= 256 mm) areas, unclassified grassland, and shrubs (including savannah, meadow, veldt, pampa, tundra) |
| 24 | 0.050 | unclassified trees, open development areas (containing parks, streets of rural character) |
| 25 | 0.055 | herbaceous wetlands |
| 26 | 0.067 | emerged barriers |
| 27 | 0.140 | hardwood woodland and cultivated woodland |
| 28 | 0.086 | unclassified wetlands (including watersheds, salt/fresh marshes, bottomland hardwood, swamps, mangrove swamps, seagrass flats, forest swamps) |
| 29 | 0.100 | forest land and unidentified forest trees evergreen forest, pasture, hay, crop, vegetation |
| 30 | 0.120 | deciduous forest, natural grassland, herbaceous lands |
| 31 | 0.150 | mixed forest, shrubs, scrub, emergent herbaceous wetlands |
| 32 | 0.240 | cultivated vegetation |
| 33 | 0.300 | unidentified densely built urbanized zones (uncharacterized structures) |
| 34 | 0.320 | very dense tall (long trunk) trees forest (jungles, etc.) |
| 35 | 0.368 | very dense and/or stiff grasslands (reedy bamboo, etc.) |
| 36 | 0.400 | very dense small forest trees and thick shrubs |

**Table 3.** Matching of Table 2's A/A for Manning coefficient list to Corine Land Cover (CLC) data.

| A/A | CLC Code | Description of CLC Label Areas' Characteristics |
|---|---|---|
| 4–8 | 111, 112 | Continuous urban fabric, Discontinuous urban fabric |
| 10–8 | 121 | Industrial or commercial units |
| 5–7 | 122 | Road and rail networks and associated land |
| 4–2 | 123 | Port areas |
| 4–5 | 124 | Airports |
| 3 | 131 | Mineral extraction sites |
| 6–4 | 132, 133, 141 | Dump sites, Construction sites, Green urban areas |
| 4–7 | 142 | Sport and leisure facilities |
| 23–14 | 211, 212 | Non-irrigated arable land, Permanently irrigated land |
| 14 | 213 | Rice fields |
| 22 | 221 | Vineyards |
| 30 | 222 | Fruit trees and berry plantations |
| 29–30 | 223 | Olive groves |
| 12–14 | 231 | Pastures |
| 27 | 241 | Annual crops associated with permanent crops |
| 27–32 | 242 | Complex cultivation patterns |
| 21–29 | 243 | Land principally occupied by agriculture, with areas of natural vegetation |
| 29–32 | 244 | Agroforestry areas |
| 29–34 | 311 | Broad-leaved forest |
| 30–34 | 312 | Coniferous forest |
| 31–34 | 313 | Mixed forest |
| 19–30 | 321 | Natural grasslands |
| 22–30 | 322 | Moors and heathland |
| 32 | 323 | Sclerophyllous vegetation |
| 31 | 324 | Transitional woodland-shrub |

| 16–15 | 331 | Beaches, dunes, sands |
|---|---|---|
| 12–9 | 332 | Bare rocks |
| 29–32 | 333 | Sparsely vegetated areas |
| 10 | 334, 335 | Burnt areas, Glaciers and perpetual snow |
| 28 | 411, 412, 421, 422 | Inland marshes, Peat bogs, Salt marshes, Salines |
| 1–16 | 423 | Intertidal flats |
| 6 | 511, 512, 521, 522 | Water courses, Water bodies, Coastal lagoons, Estuaries |
| 1 | 523 | Sea water |

*2.6. Input Data: Boundary and Initial Conditions, Simulation Time Limit*

A basic assumption of the CoastFLOOD approach, except for the steady state forcing of the flood flow on the coastal boundary with smoothly varying sea level maxima, is the non-treatment of the floodwater ebbing phenomenon. The model considers the spatiotemporally local wetting and drying of individual cells during the numerical solution, yet the computations are ceased when floodwater reaches the farthest area from the coastline or the waterfront. Thus, the model is not allowed to simulate the large-scale drying phase of floodwater receding back to the sea after the storm surge begins to decrease on the marine coastal boundary.

The application of a flood inundation model to a specific coastal area requires the definition of boundary conditions (mainly shoreline sea level and optionally onshore currents), topographic features (land elevation), and local flow resistance (bottom friction) as model parameters that control the flow characteristics. If the SSH on the coastline exceeds the MSL, then Equations (4) and (7) or (8) are activated with a value of $h(t) \equiv \text{SSH}(t)$ on the seaside boundary (ghost) cell, used to calculate the initial volume flux to all adjacent shoreland cells and then onto the floodplain cells. This implies that CoastFLOOD is driven by a Dirichlet-type boundary condition referring to local values of $h = \text{SSH} - z$ (where $z$ is the land elevation of a raster grid cell) [18], i.e., even for sea level timeseries SSH($t$) varying in the tidal cycle on the seaward side of the computational domain [17]. These conditions should last for at least a few hours and up to 3 days, given that the storm-induced sea level does not abruptly change in time but follows the slow smooth variation of the tidal constituent. Furthermore, this approach is ideal for particular scenarios of long-term MSLR or Total Water Level (TWL) on the coastline [104,105].

Although this approach actually ignores the momentum exchange effects between neighbouring cells in the floodplain and therefore introduces a restricted physical interpretation of the flow characteristics, it can capture all of the dominant features of the shallow seawater onshore flow, which leads to the rather slow propagation process (thus, seawater flux may be neglected) of coastal inundation [26,30,57,83,94]. To include the barotropic current's effect on the momentum flux of the first land cell adjacent to the seawater cell, we added an impromptu $Q_{xs} = U_{cx} \cdot dy \cdot h_{flowx}$ (similar to $Q_{ys}$; where $U_c$ is the storm surge-induced current velocity decoupled in Cartesian components $U_{cx}$ and $U_{cy}$) added to the calculated $Q_x$, $Q_y$ of Equations (7) and (9) or (10), only for the "first" dry shoreland cell. Its inclusion does not seem to drastically influence the inland flood inundation extent, but it is a step towards improvement of the physical representation of onshore seawater flow.

The storm tide (integrating surge and tide) levels can be extracted either from ocean modelling (Section 2.6.1) or from tide-gauge recordings and satellite altimetry (Section 2.6.2). The seawater elevation input can be entered as a boundary condition, representing the land-sea interface, on any cell in the computational domain.

2.6.1. Coupling with a Storm Surge Model

We coupled CoastFLOOD with the operational forecast model HiReSS, which simulates storm surges at both regional and local scales [17,19,21, 106]. The latter is a 2-D horizontal SWE hydrodynamic circulation model for the simulation of sea level variations and

depth-averaged currents, applied in large regional marine bodies and marginal seas [9,20,22–24], including several combined processes, such as:

- barotropic circulation hydrodynamics by momentum conservation and continuity SWEs;
- inverse barometer effect, i.e., the response of sea level to the atmospheric pressure gradient of large barometric systems;
- shear stresses of wind on the sea surface;
- Earth gravity and geostrophic effects (Coriolis force);
- interaction of surge-driven sea level and astronomical tides by a static model [107] approach based on the equilibrium theory of tides [108];
- ocean bottom friction;
- turbulence of horizontal eddies based on the eddy viscosity concept and the Smagorinsky model approach;
- interaction with coastal wave-induced currents by incorporating radiation stress terms in nearshore surf zones;

The model has been applied in operational forecast mode for short-term marine weather predictions and has been thoroughly validated, during the past 15 years, in the Mediterranean region against field data from in situ tide-gauge observations of storm-induced episodic SSH due to severe weather conditions or the derived Sea Level Anomaly (SLA; SLA = SSH–MSL) in inter-annual tidal cycles [2,17,20,21]. Its climatic mode counterpart, MeCSS, has also been evaluated for long-term historical simulations of mean and extreme storm surge patterns in the Mediterranean basin during (>30-year) reference periods [9,18,22–24,104,105]. Furthermore, the HiReSS model is the official numerical tool of the Operational Forecast Platform (OFP) Wave4Us, recently incorporated into the METEO.GR node managed by the National Observatory of Athens [109–111]. It is also advocated on a global scale by the Accu-Waves OFP [112] over several regional and marginal seas (e.g., Red Sea, Yellow Sea, Black Sea, Java Sea, NW Atlantic Ocean, etc.), gulfs, straits, and local aquatic bodies (e.g., Gulf of Finland, Osaka Gulf, Tokyo Gulf, Persian Gulf, English Channel, etc.), producing sea level forecasts for safer navigation in 50 important ports around the globe [113].

### 2.6.2. Boundary Conditions from Sea Level Observations

The model can also be forced by sea level observations on the study areas' coastlines, which are represented in the computational domain by marginal dry cells of the model grids during Still Water Level (SWL) conditions. Observations can be derived either from satellite altimetry (SLA) of the Copernicus Marine Service (CMS) that covers the last 30-years [114] or tide-gauges, located along the coastline [115]. The spatial resolution of the CMS product is 1/8° (~13 km), and it is provided in a daily step while the initial coverage of the dataset extends over all European seas. The Level L4 data are produced by merging observations of Topex/Poseidon, ERS1/2, Jason 1-2-3, Sentinel (3A/B and 6A), HaiYang-2A/B, Saral[-DP]/Altika, Cryosat-2, ENVISAT, and GFO altimetry missions. These satellite SLA fields have been previously used to evaluate the sea level variability in the Mediterranean Sea [116,117]. The tide-gauge observations can be derived from the Intergovernmental Oceanographic Commission (IOC) system. The IOC field data [115] have higher temporal resolution (e.g., 10-min step), but their spatial coverage along the coastline is coarser than the satellite or modelling data based on the locations of the tide-gauges, which are usually operated inside ports. Here, we focus on CoastFLOOD simulations forced by satellite-derived SLA data (see Section 4.2).

The CoastFLOOD simulations provide tide/surge-induced flooded areas due to satellite recordings or realistically modelled values of daily SLA or SSH values, respectively. From these, the timeseries' maxima are extracted, $SLA_{max}$ or $SSH_{max}$, and are separately simulated together with several extreme case scenarios of onshore TWL, typical of the east-central Mediterranean and the Greek coastal zone; i.e., 0.5, 1, 1.5, and 2 m [23,104,105].

The latter produce reference values of Flooded Areas (*FA*) in regions prone to coastal inundation, assisting in the normalization of flood extents in different case studies.

## 3. Case Studies and Data for Model Validation

### 3.1. Case Study Areas

The CoastFLOOD model was tested at 10 selected case study areas of the western Greek coastal zone (Figure 4), which are rather frequently inundated by storm surges of the Ionian Sea. Similar to tropical storms, peculiar low-pressure atmospheric systems may form in the western and central Mediterranean (namely Medicanes) and propagate from the westernmost cyclogenesis centers of the basin towards the Ionian and Adriatic Seas, making landfall on the western shores of the Italian and Balkan Peninsulas [118–121]. These events are known to threaten the selected case study areas, located in coastal lowland regions prone to inundation (Figure 4a). Thus, the latter were chosen based on a series of recorded coastal (and/or compound) flooding events that were recently reported in mass media (i.e., some examples out of numerous documented flood inundation impacts in provincial and metropolitan Greek areas; Figure 4b–e):

- Manolada-Lechaina coastal zone (Area 1), east of Patra city, north-western Peloponnese, southeastern Ionian Sea, recorded during October 2021 storm Ballos [122] followed by incidents during December of the same year (December 2021).
- Vassiliki Bay (Area 2; Figure 4c) on the southern coast of Lefkada Island, northern Ionian Sea, recorded on 17 November 2017 [123] and on 18 September 2020 [124].
- Preveza coast (Area 3; Figure 4e), west-central Epirus, northern Ionian Sea, recorded on 30 November 2021 [125].
- Igoumenitsa port (Area 4), north-western Epirus (north Ionian Sea), recorded on 12 November 2017 [126].
- Cephalonia Island (including the Livadi coastal area in its southern bay; Area 5), central Ionian Sea, recorded on 18 September 2020 [127] during Ianos Medicane [17,120,121].
- Patra city (Area 10), broader metropolitan area in north-eastern Peloponnese, Rio town's flooded seafront during Ianos Medicane [17,120,121], on 18 September 2020 [128].

Other interesting flood-prone areas (Figure 4a) frequently impacted by sea level elevation on the Ionian coastline comprise the towns of Kalamata (Messenia, southern Peloponnese; Area 6) and Argostoli (east Cephalonia Island, Ionian Sea; Area 7), the rural areas of Kyparissia (north-western Messenia, south-western Peloponnese; Area 8), and Laganas (southern Zakynthos Island, Ionian Sea; Area 9).

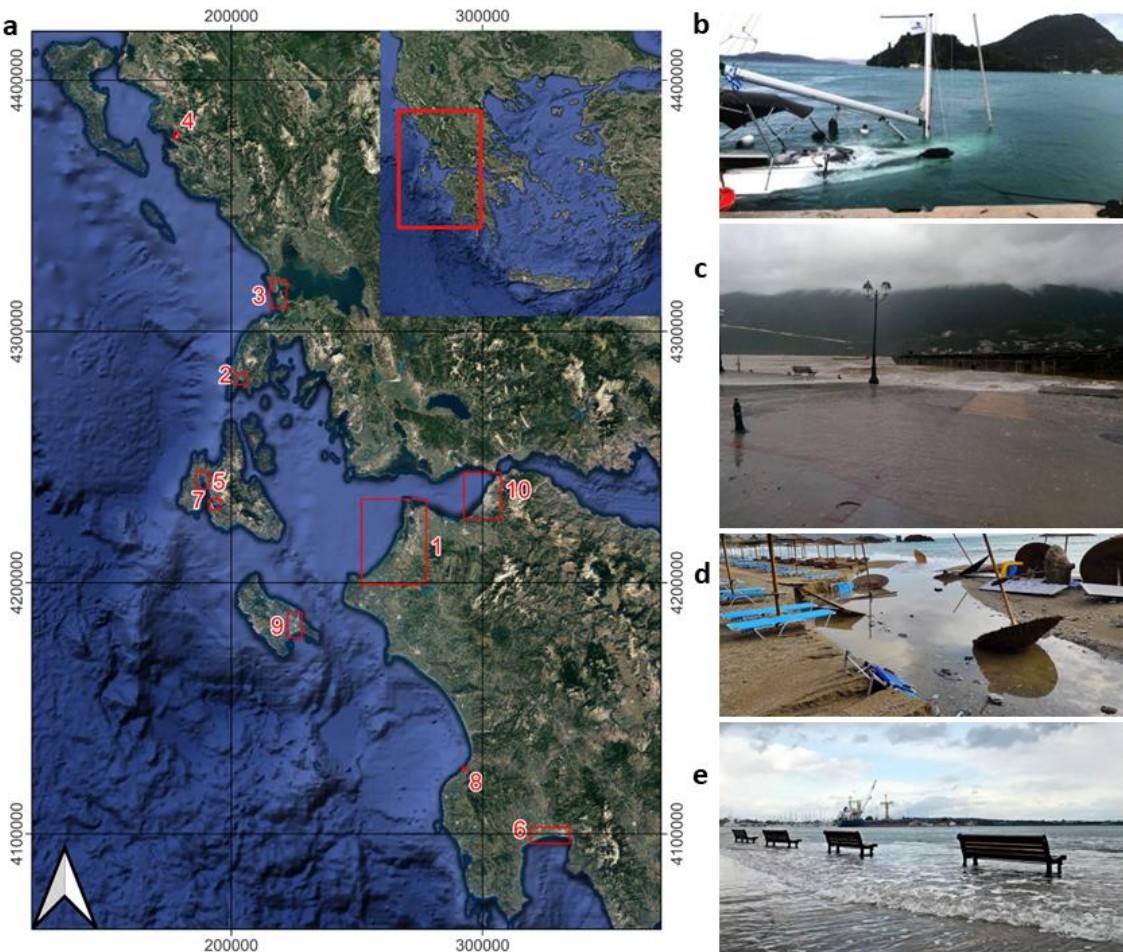

**Figure 4.** (**a**) Map of selected study areas to apply the CoastFLOOD model; Areas 1: Manolada-Lechaina, 2: Vassiliki bay, 3: Preveza coastal area, 4: Igoumenitsa port, 5: Livadi bay, 6: Kalamata, 7: Argostoli, 8: Kyparissia, 9: Laganas, 10: Patra city; (**b**) Depiction of boat wreck due to the passage of Ianos Medicane (September 2020) over Lefkada Island; (**c**) Storm seawater inundation of November 2017 in Area 2 (Vassiliki, Lefkada Island); (**d**) impact of "Ballos" (October 2021) storm on a touristic beach on Corfu Island; (**e**) storm surge coastal inundation at the seafront of Preveza (in November 2021).

### 3.2. Observational Data for Model Evaluation

The coastal model validation was based on comparisons of simulation results against ocean colour images collected by the Sentinel-2 satellite with a spatial resolution of 10 m freely distributed by the Copernicus Data Space Ecosystem (CDSE) or Sentinel Hub [129,130]. To estimate the observed coastal inundation during stormy conditions, a remote sensing technique of Sentinel-2 raster images was used to compute the NDWI [131] on coastal areas affected by storm surges, shown to oversee any alterations in water content on the Earth's surface aquatic resources [132]. Several researchers have used NDWI in the past to assess flood extents due to hurricane-led storm surges, e.g., in the Gulf of Mexico, or to identify coastlines [133,134]. NDWI is computed based on *Band*3 and *Band*8 bands of the ocean colour images:

$$NDWI = \frac{Band3 - Band8}{Band3 + Band8} \tag{14}$$

where *Band*3 is the Visible Green Light (VGL) and *Band*8 is the Near-Infrared Radiation (NIR) of the spectrum.

Herein, we use NDWI to identify (wet) flooded areas on low coastal inlands following a storm surge event with two different procedures (see Section 4). In the first case, a

satellite image taken on 14 December 2021 was used to calculate the NDWI > 0 on the raster grid, corresponding to "wet" cells of the study area (given that they are not all flooded by seawater, but by rainwater from precipitation or surface runoff as well). The second approach involved the estimation of flooded areas using two separate satellite images, the one before (15 September 2020) and the second after (20 September 2020) the recorded Ianos Medicane's storm surge that occurred on 17–18 September 2020 (due to the unavailability of datasets on these exact dates); the difference of the two calculated NDWIs was used to estimate the inundated area (Androulidakis et al., 2023). Specifically, after calculating the individual NDWI for each cell of the domain and both images, the difference in $NDWI_{dif}$ of post-storm NDWI minus pre-storm NDWI was calculated on each pixel of the raster grid. It was assumed that pixels with $|NDWI_{dif}| > 0.15$ corresponded to remaining wet ground (areas that were very likely flooded during the storm). To identify areas that were flooded likely due to the storm surge, the NDWI values of the second image were filtered to exclude the already wet cells before the storm surge. Notably, the areas identified as inundated by stormwater had $NDWI_{dif} > 0.5$ in many instances, confirming the result. To avoid misinterpretations, we mainly considered lowland areas close to the coastline (with hydraulic connectivity to the sea), nevertheless there is no safe method yet able to distinguish the source of floodwater (e.g., tidal surge, drainage or runoff, and rainfall) based on the NDWI technique.

An important limitation of the comparison with remote sensing NDWI fields is that satellite images are susceptible to the timeframe they refer to or are available in, namely due to the absence of satellites over the study regions during the storm event or due to cloud contamination, a process very common during storms, cyclones, and Medicanes. A second limitation of the NDWI method is that the water accumulation due to intense water precipitation or surface runoff from surrounding higher ground into bilged lowlands (e.g., cesspools, dugouts, sumps, pits, fosses, and cisterns) can contaminate the derived NDWI fields of humid surfaces or wetted areas, thus deregulating the coastal model validation procedure. Nevertheless, the NDWI method is essential for model performance testing of the occurrence of characteristic coastal hazard events.

### 3.3. Enhanced Bathtub Module for Model Validation

The CoastFLOOD model was compared with a static level "bathtub" approach inundation module [78,135]. This method easily identifies the flood-prone low-lying areas with ground elevation below a predefined threshold, e.g., an estimation of coastal seawater level maximum, $z < SSH$ or $z < TWL$. The bathtub technique is known to be oversimplifying in terms of physical processes and can produce serious overestimations of coastal flood extents [34,136]. Therefore, an enhanced bathtub module with hydraulic connectivity (Bathtub-HC) was adopted [81,137,138]. To this end, we applied a nearest neighbour search algorithm following the 'eight-side rule' in order to identify the potential floodwater flow path between neighbouring raster-grid cells in both cardinal (cross-orthogonal) and ordinal (diagonal) directions of the horizon. This way, the unsubstantial excessive estimations of possible seawater inundation in coastal lowlands was restricted.

The Bathtub-HC method is known to provide fast and adequately robust estimations of flooded coastal area extents, yet they are practically more conservative than those by SWE models. Compared to the CoastFLOOD model, this method neglects the floodplain terrain sloping topography, the bottom friction effects, etc. Thus, it can predict the flooded areas, but it cannot account for flood duration, detailed floodwater height, and fluxes (velocities) that dynamically affect the onshore and overland floodwater flow. Hence, the Bathtub-HC results are usually only implemented as a reference level for potentially wet inland cells in evaluative assessments of reduced complexity numerical models [16–18]. Moreover, bathtub methods can collaterally identify and depict lowland bilge areas (e.g., pits, fosses, puddles, and cisterns) that can accumulate water from rainfall and surface runoff, unlike SWE coastal flooding models, which only account for seawater floods [17].

Convenient field data of coastal inundation based on in situ observations of floodwater height and extents are literally very rare, while their fitness for model verification is not always suitable due to several reasons [30]. There are no available floodwater level gauges in coastal areas (there are only a very few downstream of river embankments in fluvial floodplains), at least in Greek (scarce- or no-data) study areas of interest. Reduced complexity models of coastal flooding need field data for verification on geophysical scales (10–100 km wide) of observation and implementation. Therefore, only satellite data can serve as in situ references for impacted areas due to seawater flooding. The latter are susceptible to the timeframe they may be available in (e.g., the absence of satellite data during storm events, cloud contamination of satellite images, etc.). Uncertainty regarding the contribution of possible sources of recorded inundation besides storm surges (e.g., waves, rainfall, drainage) may obfuscate the derivation of inundated area coverage due to storm-induced floods.

## 4. Results

We examined the adequacy of the inundation model predictions under realistic severe storm surge conditions (Section 4.1) and simplified bathtub estimations (Section 4.2), on the Ionian Sea coasts of Greece. Idealized (extreme) and realistic (maxima from 2017–2021 period) scenarios of coastal flooding are also presented in Section 4.3.

### 4.1. Model Verification against Satellite Data during Severe Storm Surge Conditions

Two areas and events were used for qualitative verification of the coastal flooding model's performance due to a lack of imperative satellite data. In the Manolada-Lechaina study area, there was unfortunately no satellite data availability during October 2021, when storm Ballos hit. However, NDWI could be estimated on 14 December 2021, when another storm surge incident was traced based on the retrieved SLA datasets. These depictions serve as reference for qualitative comparisons with the modelled output of CoastFLOOD. Figures 5 and 6 present flood maps of model simulations overlaid by satellite-tracked wet regions. The CoastFLOOD results are driven on the coastal boundary of the Manolada-Lechaina study area by recorded SLA values on 14 December 2021 (see Section 2.6.2). The zoomed-in maps of Figure 6 depict the overlap of NDWI-identified wet areas by satellite images above flood inundation model output focusing on the mainly affected northern and southern parts of the study area. In general, the CoastFLOOD simulations seem to reproduce the coastal flooding mechanism in areas that are more-or-less affected (wetted) by stormy weather during the timeframe of analysis. Furthermore, model results may overpredict the momentary depiction of flood extents, as derived by the NDWI method based on the recorded image on 14 December 2021 at 09:24:01 (hh:mm:ss). However, there is no guarantee that the satellite data represent the actual situation of floodwater extents during the storm-induced high seas.

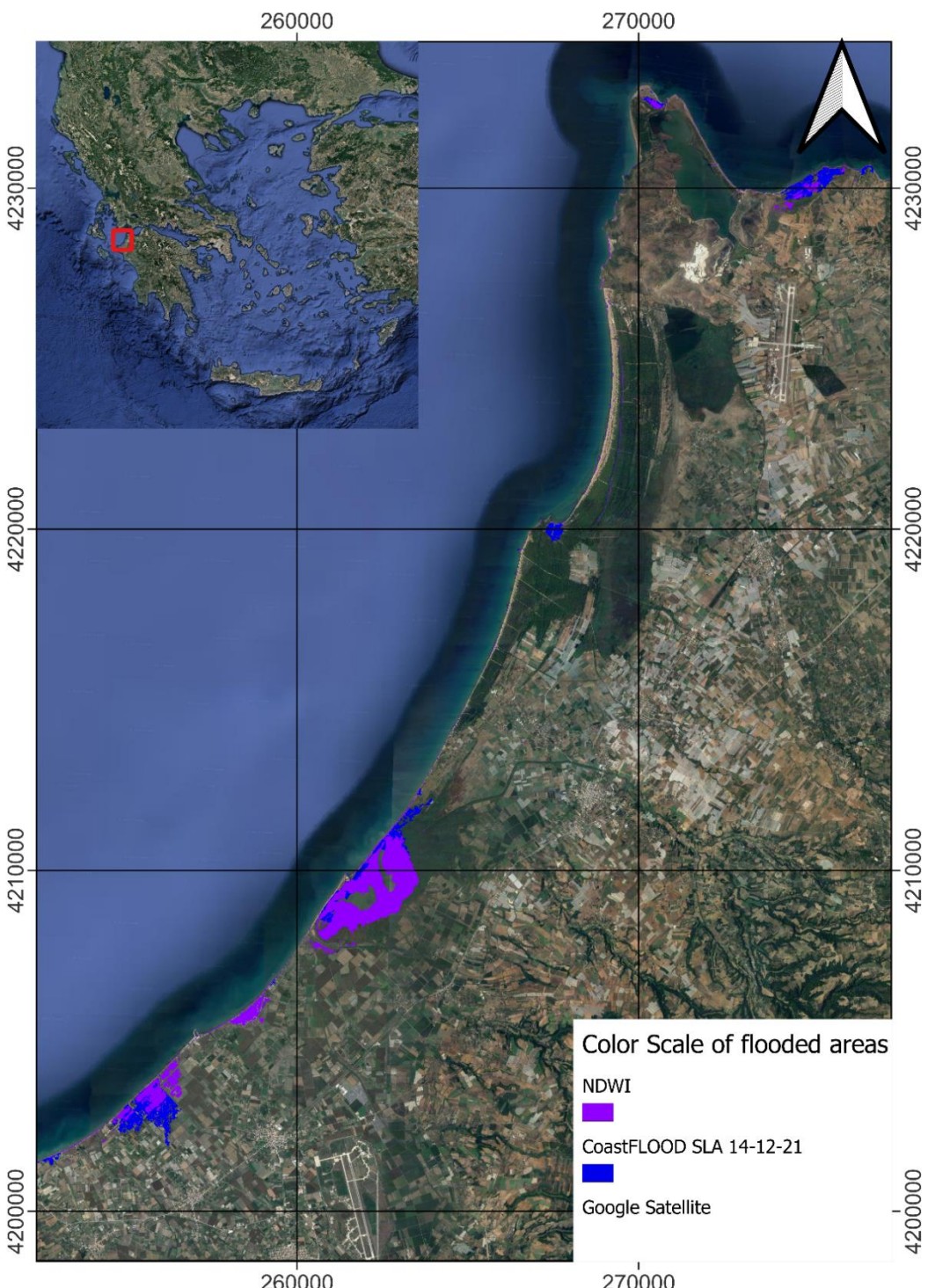

**Figure 5.** Map of estimated flooded areas as depicted by NDWI satellite data (purple colour) over-laid on CoastFLOOD simulation results driven by recorded SLA values on 14 December 2021 (blue colour) for the Manolada-Lechaina study area, north-western Peloponnese (western Greece). The flooded areas' extents are superimposed over a background of recent GoogleEarth satellite images.

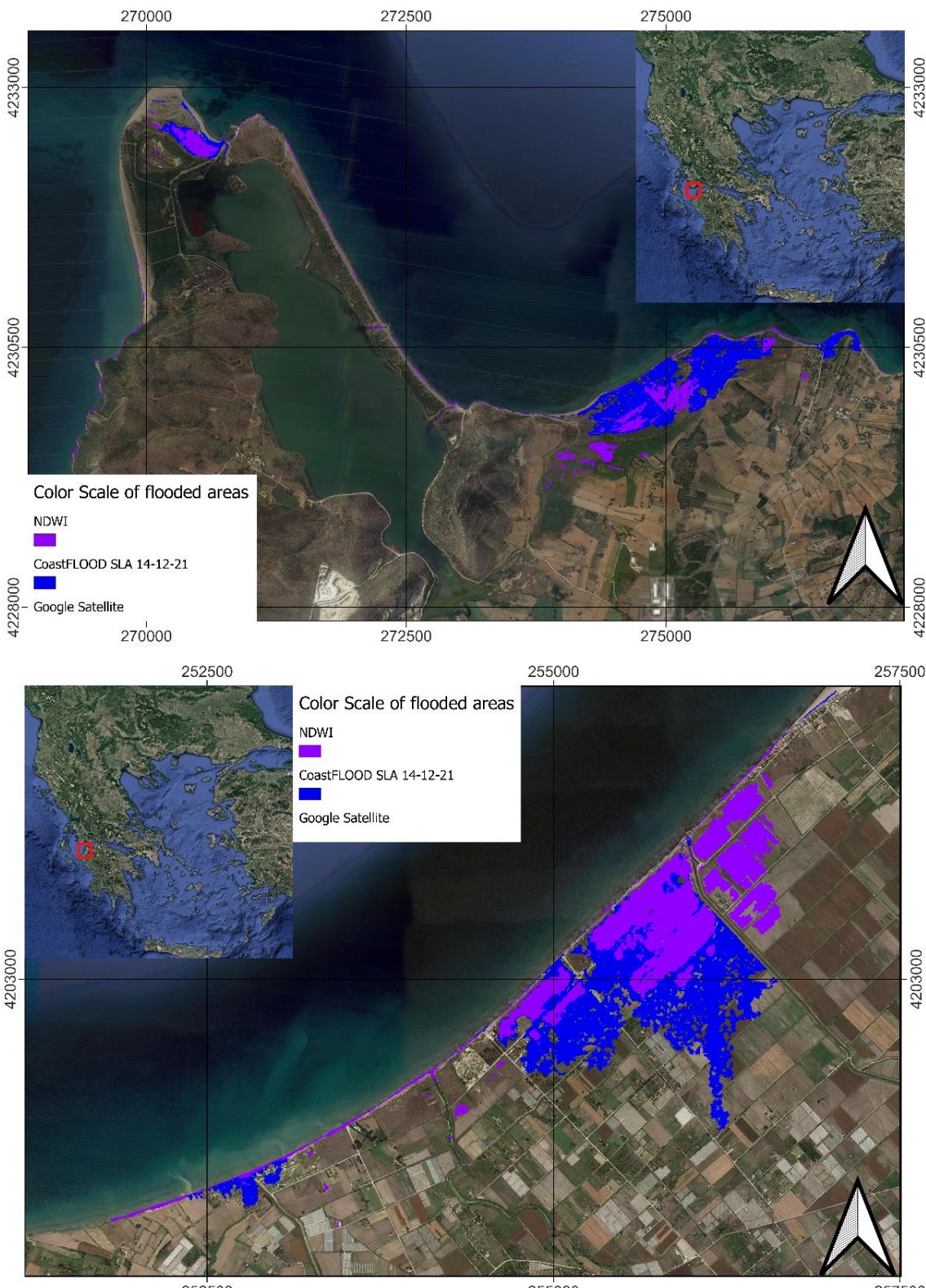

**Figure 6.** Zoom-in maps from the estimated flooded areas in Figure 5 as depicted by NDWI satellite data (purple colour) overlaid on CoastFLOOD simulation results driven by recorded SLA values on 14 December 2021 (blue colour) for the Manolada-Lechaina study area; upper map: northern part, lower map: southern part. The flooded areas' extents are superimposed over a background of recent GoogleEarth satellite images.

Figure 7 portrays the estimated flood map in the Livadi study area (Area 7; Figure 4a), based on a heuristic approach of NDWI differences before and after the landfall of Ianos Medicane on the study area [16,17,120,128], depicted on a 0.1–1.0 scale of values, overlaid on the CoastFLOOD simulation results. The latter were driven by HiReSS-modelled SSH (see Section 2.6.1) from operational forecasts by the WaveForUs system on 17 September 2020 [17,109]. The predicted flood extents (red patches) on the southern coastal zone of Cephalonia Island overlap and include the traced wet areas by remote sensing (purple patches). A large wet area (shown in shades of purple) in the northern part of the study area, set off of the model-predicted flooded area, is considered to be hydraulically detached from the impacted area due to the storm surge. These areas usually act as drainage bilges that are usually flooded with water originating from local intense rainfall and/or stormwater surface runoff from the surrounding hills and mountains. An extreme case scenario of TWL = 1 m, typical for a possible cumulative sea level increase due to the combined effects of surges and waves, is also provided (yellow patches) for comparison of the flood-prone littorals against the actually impacted touristic coastal areas. The extreme case flooded extents may reach a 250-m distance onshore in the southern part of the study area, occasionally reproduced by the model for the actually recorded $SLA_{max}$, too, while not along the entire beach stretch. An intrusion of floodwater around 167 m from the coastline, where the beach dunes are located parallel to the shoreline contour, is further plausibly reproduced by the model for both $SLA_{max}$ and TWL cases on the northwestern part of the coast. For the extreme TWL case, the model further predicts a 458-m inland flood extent on the northern part of the study area, but this is not reproduced by the $SLA_{max} = 0.262$ m simulations as the area is not hydraulically connected to the sea by an equally low land pathway.

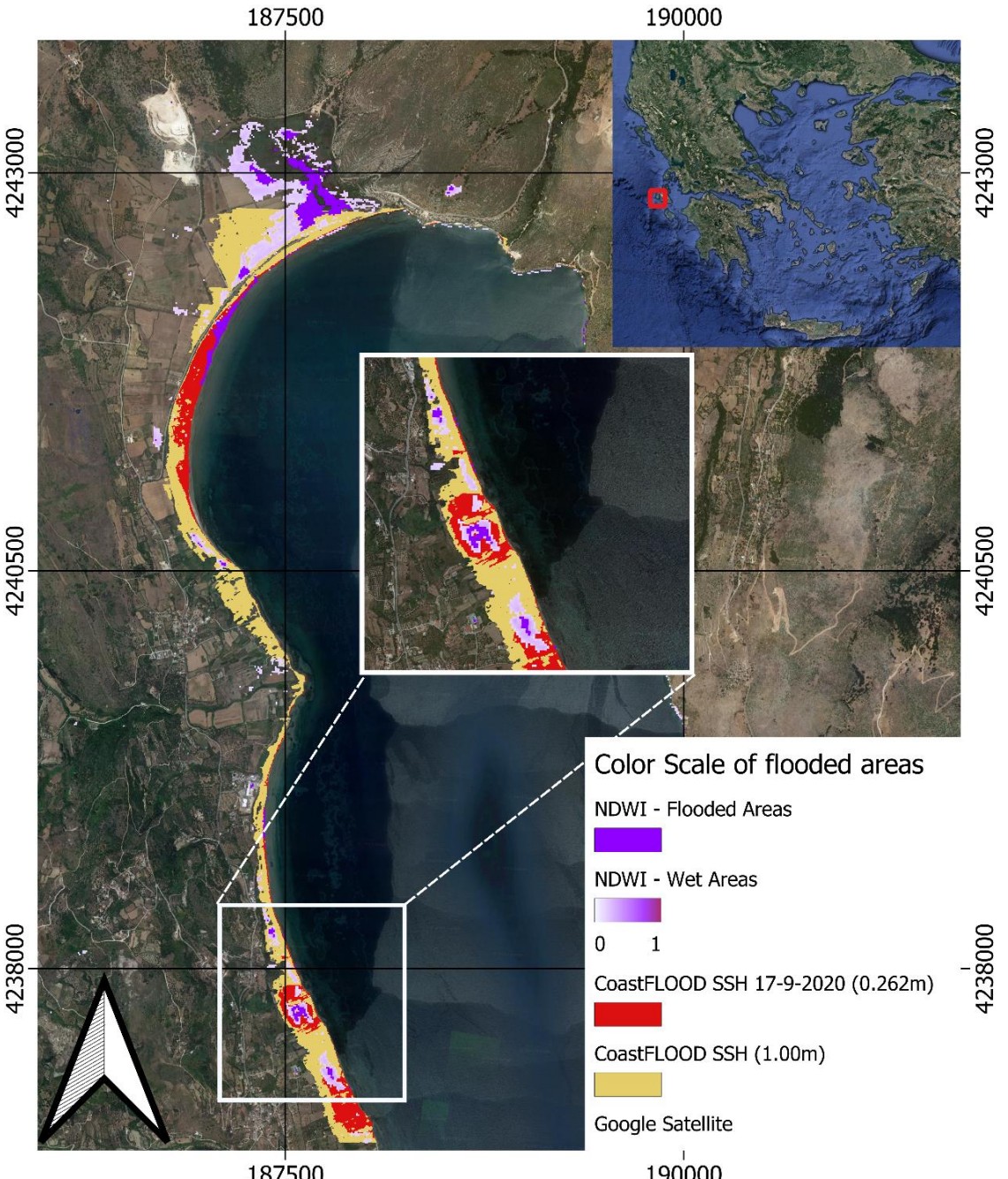

**Figure 7.** Map of estimated flooded and wet areas as depicted by NDWI differences by satellite data before and after Ianos Medicane passage from the study area (white-to-purple colour shift corresponding to 0–1 of NDWI values; method description in Section 3.2), overlaid on CoastFLOOD simulation results driven by HiReSS-modelled SSH from operational forecasts by the WaveForUs system, during the Ianos Medicane landfall on 17 September 2020 (red colour) flood extent magnitude, for the Livadi study area, on Cephalonia Island, in the Ionian Sea. Modelled flood area extents for an extreme case scenario of TWL = 1 m is also provided in yellow color. The insert map presents a zoomed-in depiction of the main impacted area corresponding to 17 September 2020, $SLA_{max}$ = 0.262 m underlaid below the identified wet areas by the NDWI methodology (white-to-purple color).

## 4.2. Model Validation against the Bathtub-HC Approach

To validate the CoastFLOOD model's efficiency to reproduce the highest possible flood extent (on the safe-side in terms of engineering) in coastal plains, we implemented the performance metric goodness-of-fit, *GoF*, between the modelled (CoastFLOOD;

subscript: mod$_{CF}$) and the estimated (Bathtub-HC; subscript: est$_{BHC}$) flooded area, *FA*, extents [26,30]:

$$GoF = \frac{FA_{mod_{CF}} \cap FA_{est_{BHC}}}{FA_{mod_{CF}} \cup FA_{est_{BHC}}} \tag{15}$$

where *FA* is defined by the amount of flooded grid cells by the CoastFLOOD model and Bathtub-HC estimations, respectively. The two predictions exactly overlap each other if *GoF* = 1 and no intersection of *FAs* occurs for *GoF* = 0 [94]. In the simulated test cases shown in Figures 8–11, the CoastFLOOD model agreement compared to Bathtub-HC was very high, i.e., *GoF* > 0.95 (see captions of Figures 8–11 for actual values), for several scenarios of SLA as a driver of coastal inundation, ranging from a recorded SLA$_{max}$ ≈ 0.25 m (minimum SLA$_{max}$ recorded in Area 1) to extreme cases of TWL = 1.0–1.5 m. The model was able to evenly reproduce the estimated maximum inundation extent over lowland areas using the bathtub approach. As expected, it was slightly underestimated compared to the latter, yet, therefore, CoastFLOOD shows a more realistic perspective of littoral inundation, given the error of the retrieved DEM/DSM topography and the boundary conditions (SLA on the coastline) provided by satellite observations.

Figures 8 and 9 present maps of flooded areas driven by storm surge maxima of SLA > 0.25 m in Kalamata (Area 6) and Manolada-Lechaina (Area 1) with a plausible, nearly perfect overlap of the two methodologies, only showing "wet-area" differences (i.e., Bathtub-HC overestimations) in inland areas far away from the coastal boundary. Similar flood model behaviour is observed for an extreme case scenario of TWL = 1.5 m in one of these study areas. Figures 10 and 11 present maps of flooded areas driven by storm tide extremes of TWL ≥ 1 m in Preveza (Area 3) and Argostoli (Area 7), with an equally persuasive overlap of the two methodological results. The inland flood-prone areas identified through the Bathtub-HC approach are obviously located in inclined higher grounds on the maximum boundary of the floodwater extent modelled with CoastFLOOD. The model achieves similar performance in both the natural and urban settings.

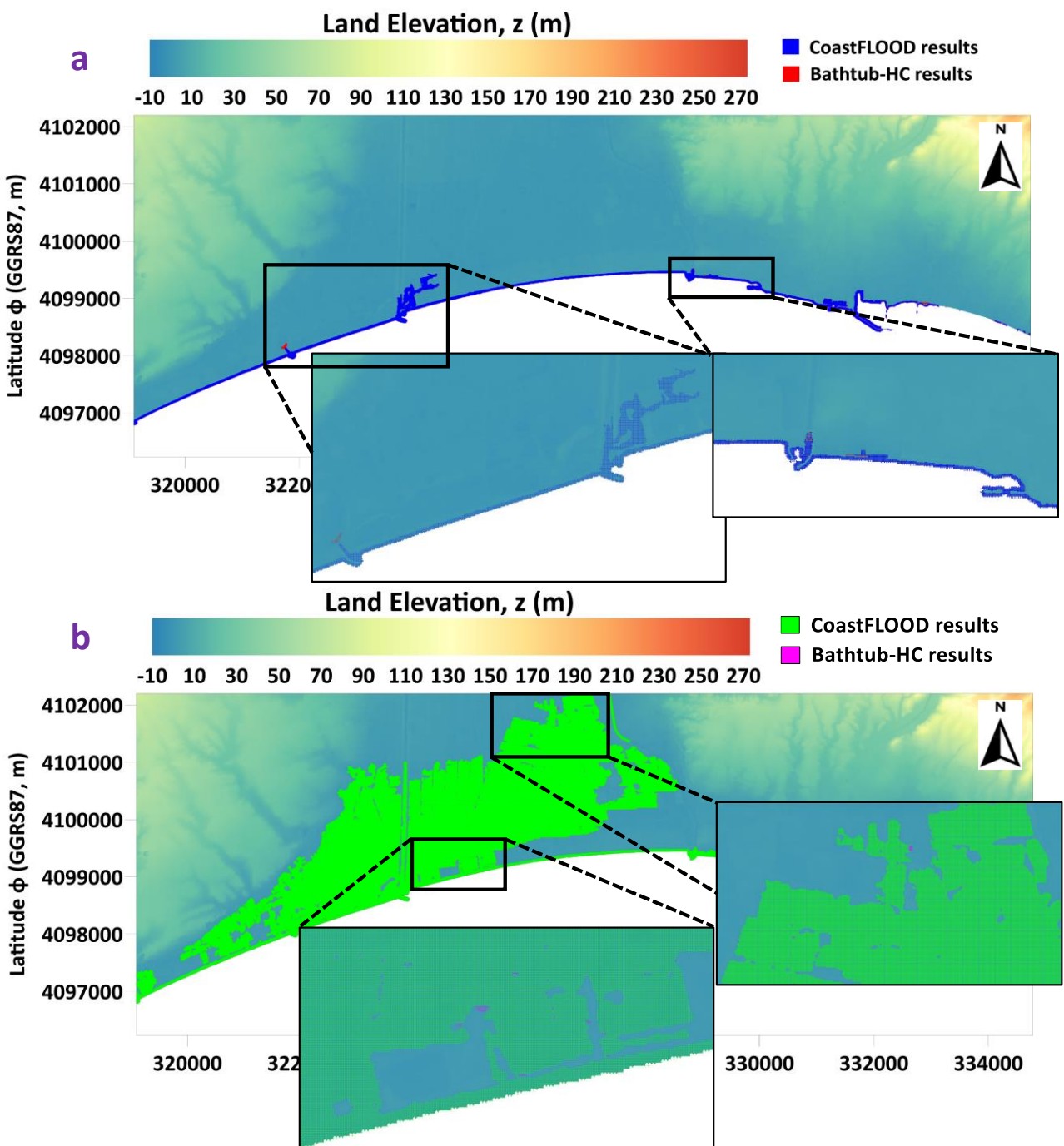

**Figure 8.** (**a**) Maps of estimated flooded areas as depicted by Bathtub-HC approach (red colour) and CoastFLOOD simulations (blue colour), driven by $SLA_{max} = 0.253$ m during December 2021, for the Kalamata coastal zone (Area 6), in Messenia of the southern Peloponnese. The insert maps present zoomed-in depictions of the main impacted areas showing the good agreement of the two methods and the superimposed discrepancies of flood extents on the boundaries of the floodwater "wet" regions ($GoF = 0.972$). (**b**) Maps of estimated flooded areas as depicted by Bathtub-HC approach (purple colour) and CoastFLOOD simulations (green colour), driven by an extreme scenario of TWL $= 1.5$ m, for the same study area, including respective zoom insert maps ($GoF = 0.993$). The two results overlap each other in such a way that Bathtub-HC red and purple areas are barely visible.

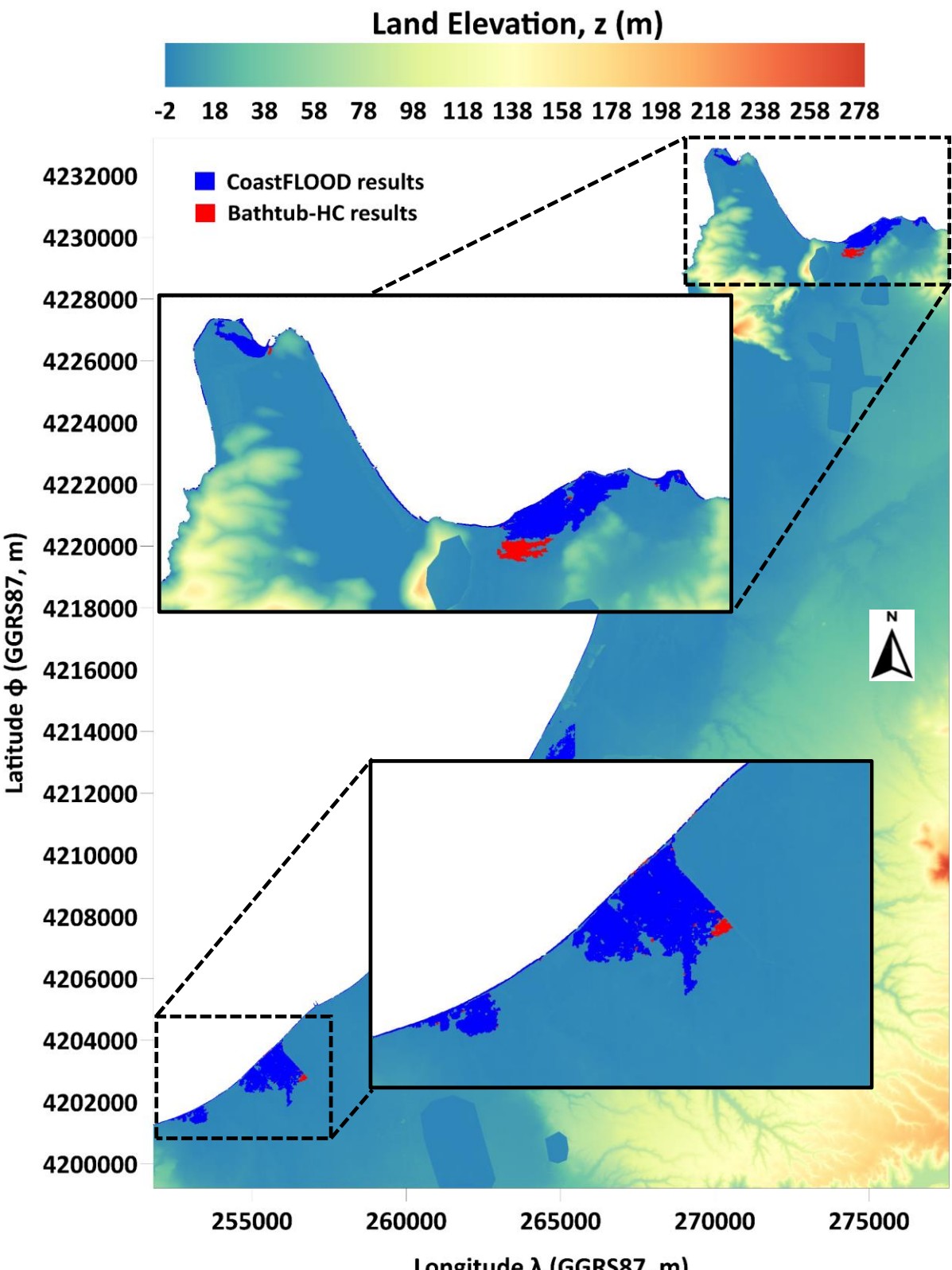

**Figure 9.** Maps of estimated flooded areas as depicted by Bathtub-HC approach (red colour) and CoastFLOOD simulations (blue colour), driven by $SLA_{max}$ = 0.25 m during December 2021, for the Manolada-Lechaina coastal zone (Area 1), in north-western Peloponnese. The insert maps present zoomed-in depictions of the main impacted areas showing the good agreement of the two methods and the superimposed discrepancies of flood extents on the boundaries of the floodwater "wet" regions (*GoF* = 0.951).

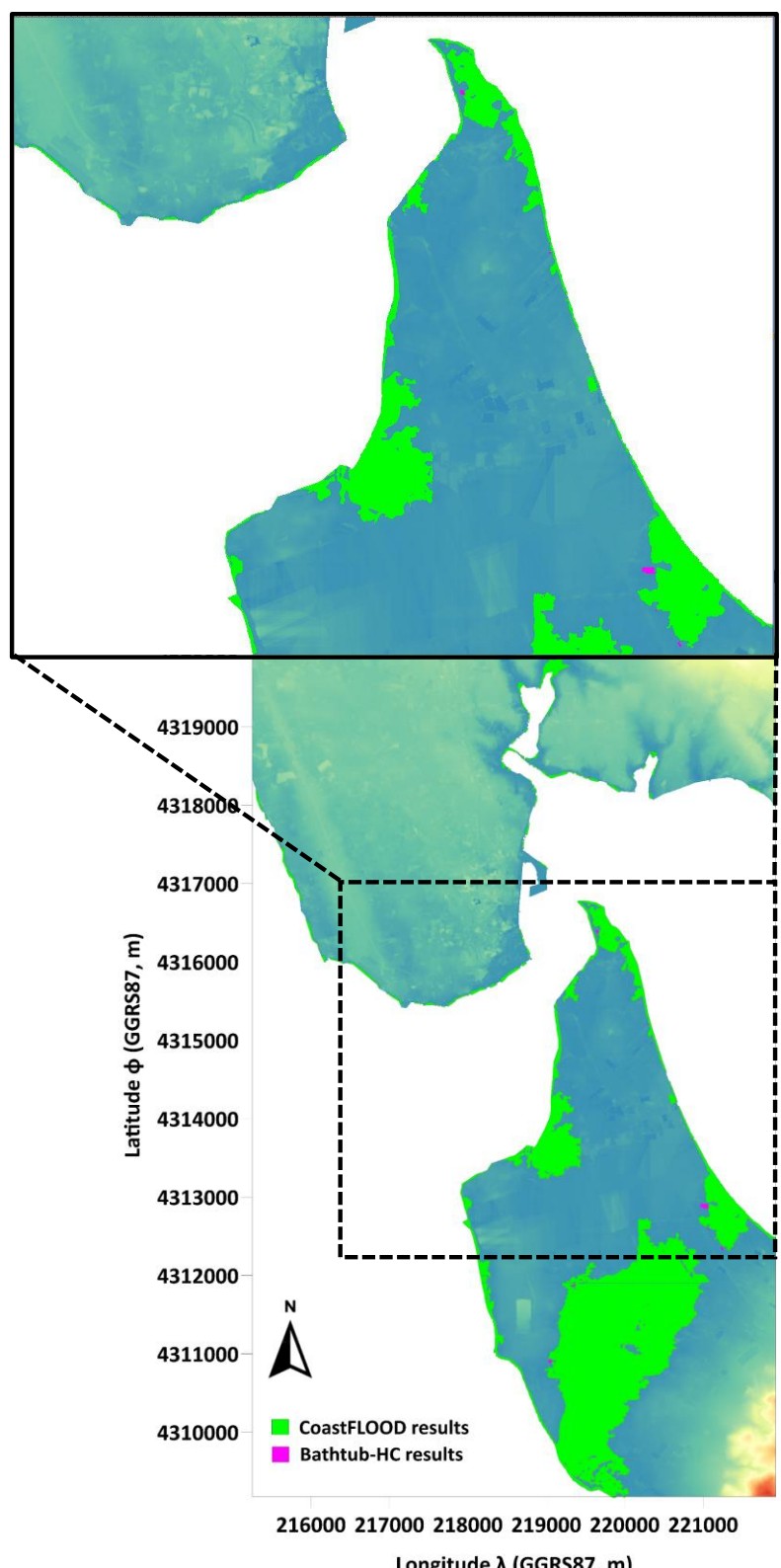

**Figure 10.** Map of estimated flooded areas as depicted by Bathtub-HC approach (purple colour) and CoastFLOOD simulations (green colour), driven by an extreme case scenario of TWL = 1.5 m, for the Preveza coastal case study (Area 3), in western Epirus. The insert map presents a zoomed-in depiction of the main impacted areas showing the good agreement of the two methods in tandem with the superimposed discrepancies of flood extents on the boundaries of the floodwater "wet" regions (*GoF* = 0.984).

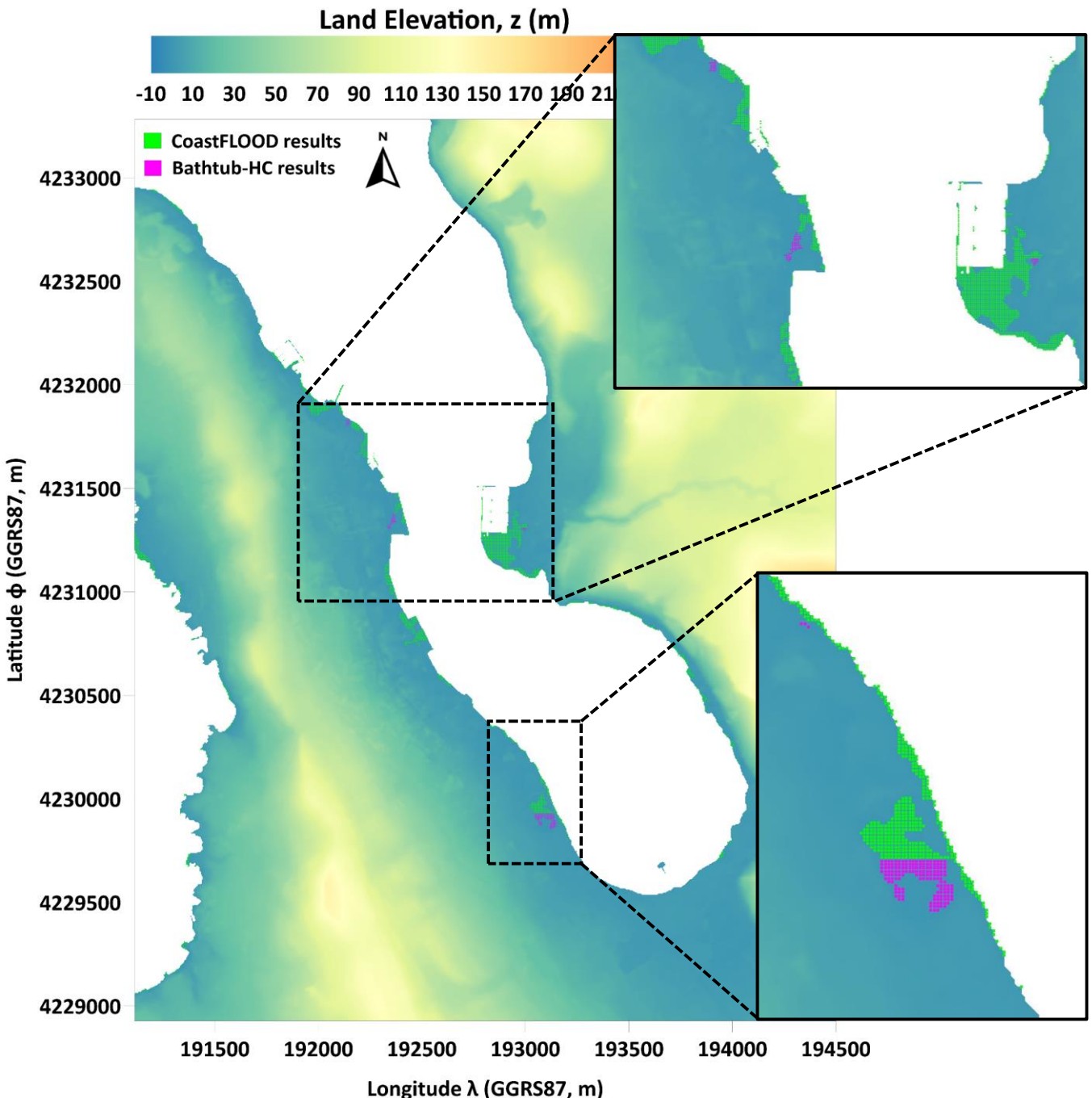

**Figure 11.** Map of estimated flooded areas as depicted by Bathtub-HC approach (purple colour) and CoastFLOOD simulations (green colour), driven by an extreme case scenario of TWL = 1 m, for the Argostoli coastal inlet (Area 7), in Cephalonia Island; the good agreement of the two methods is depicted in tandem with the superimposed discrepancies of flood extents on the boundaries of the floodwater "wet" regions (*GoF* = 0.96).

*4.3. Flooding Scenarios of Realistic and Extreme Sea Level Conditions*

Within the framework of setting up an operational modelling platform for storm surge flooding in Greece towards a robust Early Warning System for coastal hazards [51,139–143], we presented CoastFLOOD outputs in the selected study areas.

Figure 12 shows a map of operationally modelled flooded areas, driven by a mild storm surge of SLA = 0.23 m (a maximum record in the 2017–2021 period) and additional extreme case scenarios of TWL = 0.5–2.0 m, for the coastal zone of Kyparissia (Area 8; north-western Messenia, south-western Peloponnese). The impacted areas associated

with the satellite-derived sea level mainly refer to the first few tens of meters from the shoreline, which are more pronounced in the northern part of the Kyparissia coast. In general, the occurred storm surge maximum presented no serious impacts on the waterfront of the port area (marina and fishing harbour), but in the case of an extreme event (e.g., TWL ≥ 1.5 m), the flood expanse could locally reach up to 100–150 m onshore from the shoreline extending along the entire coastal stretch. In that case, the residential areas behind the port infrastructure can also be affected. The use of a global effective grid-scale Manning coefficient (*n* = 0.02 corresponding to A/A 9 of Table 2), compared to a properly distributed field of gridded *n* values based on CLC datasets in the area, does not highly affect the estimation of the flood extent and the location of impacted areas, but it drastically influences the calculation of the timespan for maximum flood reach, rendering it from almost half an hour to 49.2 min (0.82 hrs; Table 4), respectively.

**Table 4.** Timeframe for Maximum Flood Inundation Reach, $t_{MIR}$.

| SLA (m) | 0.2–0.3 | 0.5 | 1 | 1.5 | 2 |
|---|---|---|---|---|---|
| **Study Area** | $t_{MIR}$ **(hrs)** | | | | |
| Laganas | 4.25 | 3.61 | 4.45 | 6.40 | 8.87 |
| Kyparissia | 0.82 | 0.72 | 1.12 | 1.98 | 2.21 |
| Kalamata | 3.96 | 5.13 | 25.76 | 28.59 | 32.79 |
| Patra | 14.46 | 15.93 | 50.12 | 77.39 | 81.97 |
| Vassiliki | 0.18 | 0.45 | 1.11 | 4.21 | 8.90 |
| Livadi | 0.22 | 0.49 | 5.33 | 19.87 | 38.43 |
| Igoumenitsa | 0.20 | 0.32 | 0.93 | 3.76 | 5.28 |
| Argostoli | 0.67 | 1.57 | 6.97 | 9.23 | 10.18 |

\* *The two highlighted rows correspond to exceptional cases of counterintuitively higher values of* $t_{MIR}$ *for lower values of SLA = 0.2–0.3 m.*

Figure 13 presents the map of simulated flood extents, due to a recorded $SLA_{max}$ = 0.266 m and four hypothetical extreme case scenarios of TWL = 0.5–2.0 m, for the coastal study areas on Zakynthos Island (Area 9) pertaining to Laganas beach (south) and the coastal town of Zante (north), the main port of the island. The southern beach of Laganas with the small fishing harbour on its south boundary cape is mainly impacted. The affected coastal stretch expands for several km along the entire Laganas bay with a cross-shore floodwater uprush of a maximum of 500 m inland for the extreme case of TWL = 2 m. The Zakynthos seaport in the northern part of the study area does not present any crucial impacts for regular $SLA_{max}$ < 0.3 m, but in the case of extreme events (e.g., TWL ≥ 1.5 m), the leeward breakwater/jetty and parts of the secondary harbour's docks may be overtopped by high seas. The suburban coasts can be also affected by extreme sea levels, increasing the coastal flood risk for the adjacent coastal residencies.

The situation of storm surge impacts is similar for the coastal areas of Patra (Area 10), Vassiliki (Area 2), and Igoumenitsa (Area 4), presented in Figures 14–16. In the city of Patra, the town of Igoumenitsa, and their peri-urban coastal settings (Figures 14 and 16), the lowland shores can even be flooded by rather low values of storm surge maxima (e.g., SLA = 0.24–0.28 m); nonetheless, the impacts of inundation can be quite high with flood extents reaching hundreds of meters inland for the extreme cases of SLA or TWL > 1 m; i.e., by combining the tidal surge with the wave-induced run-up. In these two study areas, the urban spaces, where high-density populations and revenue-oriented assets are located, including the port-related infrastructure, open air locales, and road networks, are more exposed to surge-flood inundation. However, in Vassiliki bay (Area 2; Lefkada Island, Figure 15), the natural coastal sites and the surrounding touristic residencies may be more likely to be impacted by extreme seawater floods, rather than the small harbour in the north-eastern part of the bay.

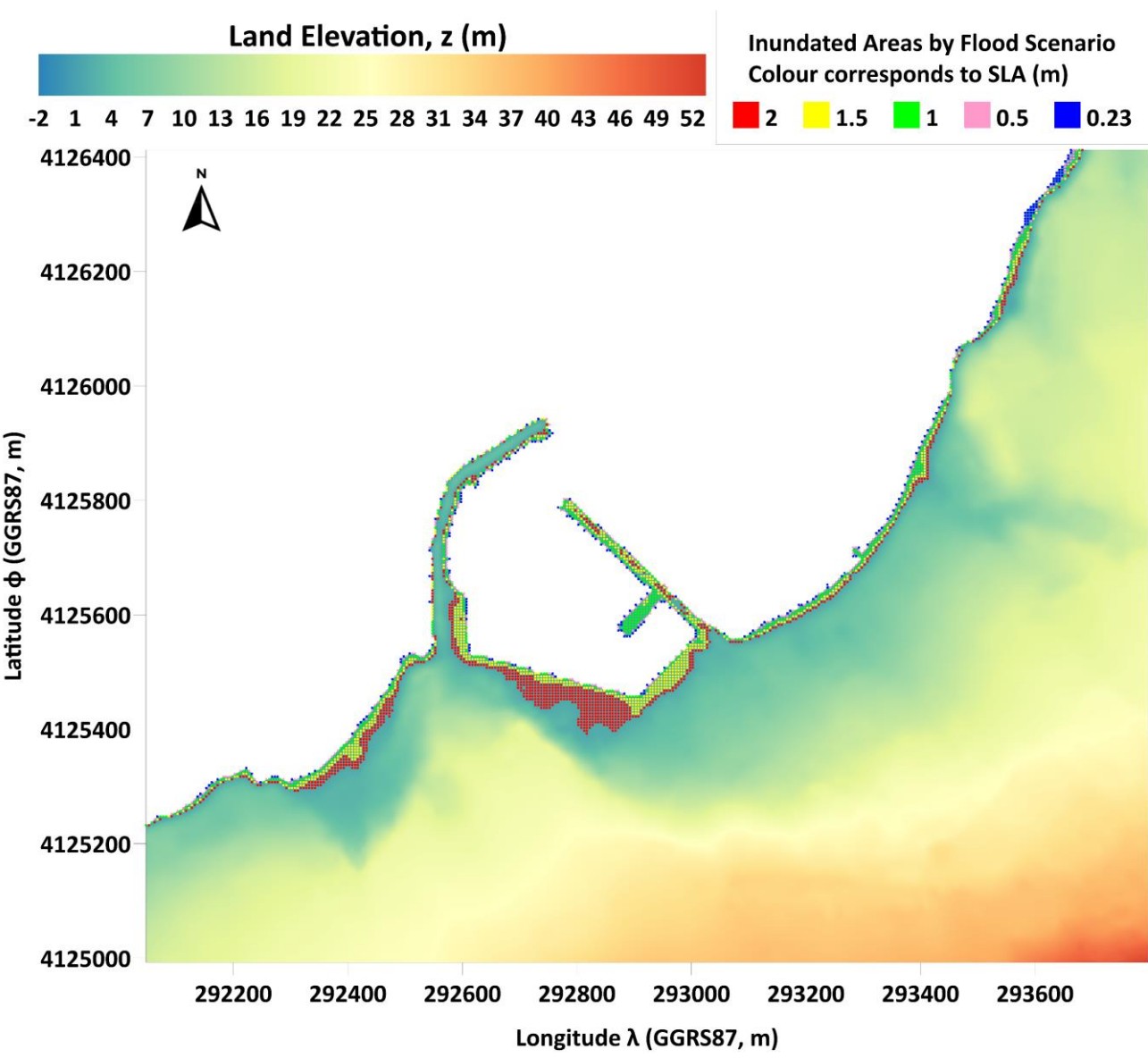

**Figure 12.** Map of estimated flooded areas as depicted by operational CoastFLOOD simulations, driven by an in situ recorded SLA = 0.23 m and four extreme case scenarios of TWL = 0.5–2.0 m, for the coastal study area of Kyparissia (Area 8; north-western Messenia, south-western Peloponnese), including a local marina harbour.

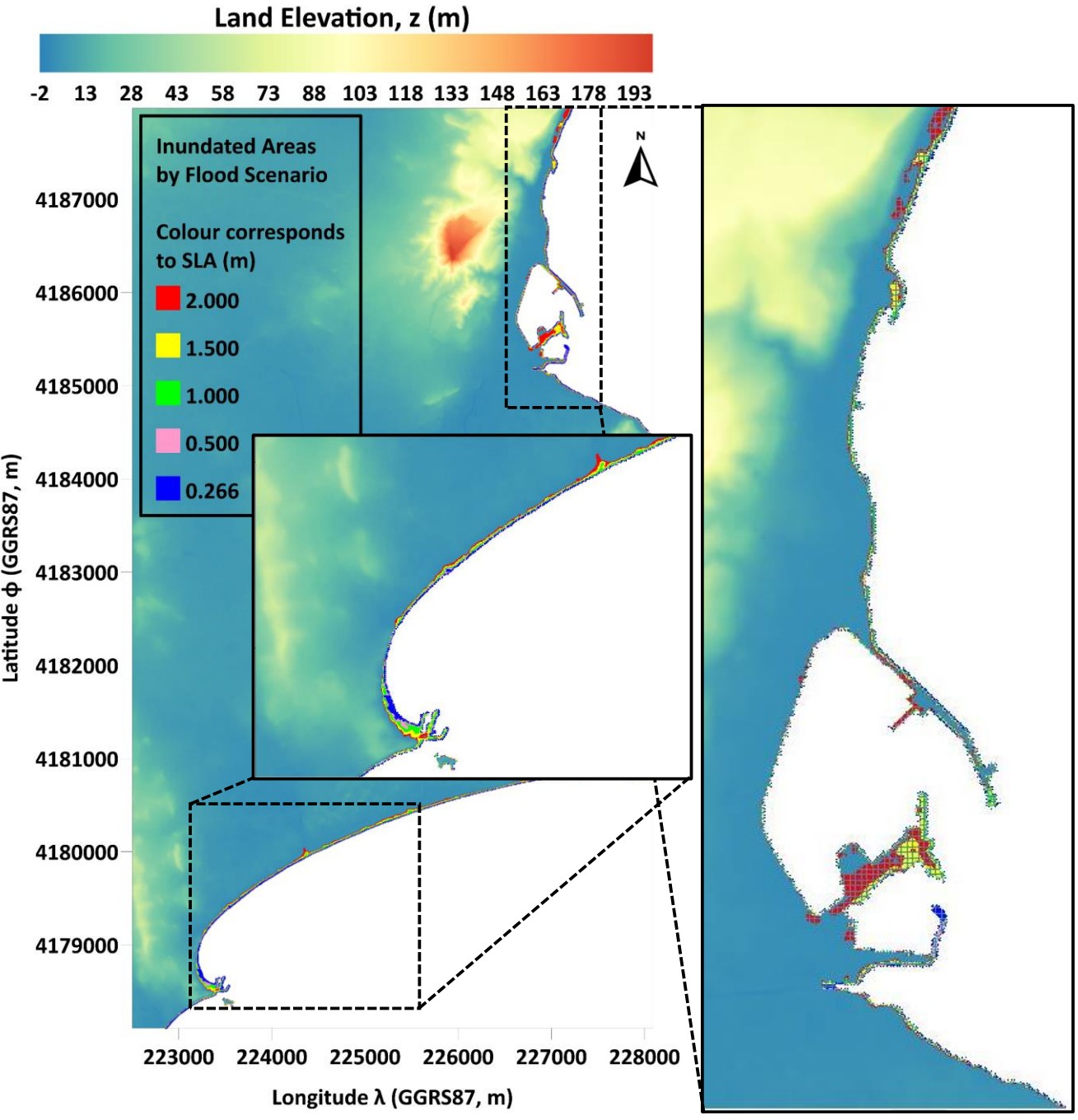

**Figure 13.** Map of estimated flooded areas as depicted by operational CoastFLOOD simulations, driven by an in situ recorded SLA = 0.266 m and four extreme case scenarios of TWL = 0.5–2.0 m, for the coastal study area of Laganas (Area 9; southern Zakynthos Island, Ionian Sea), also including Zakynthos' main port in the northern part.

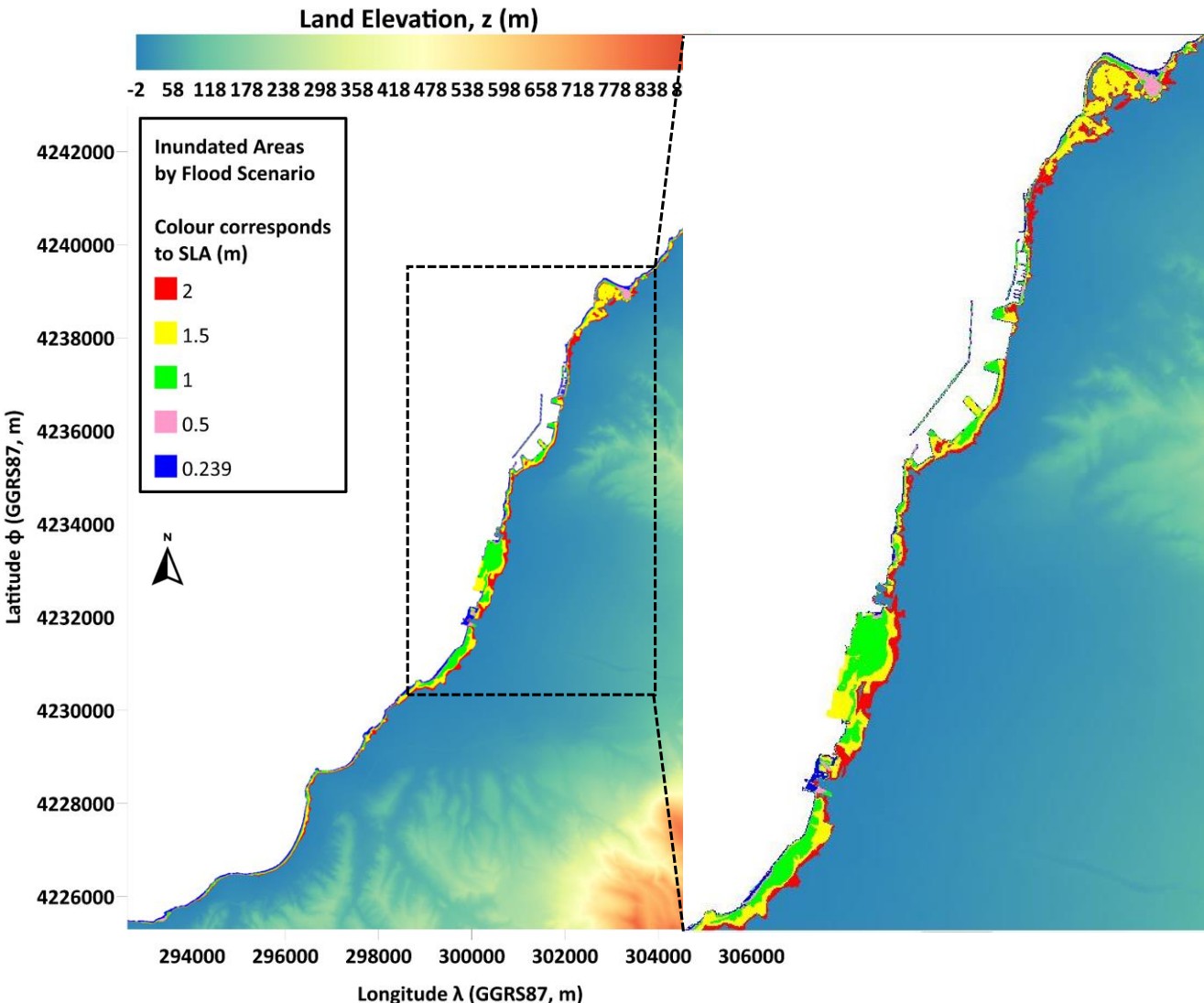

**Figure 14.** Map of estimated flooded areas as depicted by operational CoastFLOOD simulations, driven by an in situ recorded SLA = 0.239 m and four extreme case scenarios of TWL = 0.5–2.0 m, for the city of Patra (Area 10; north-eastern Peloponnese), also including the main port in the central part, the rural coastal areas of Achaia around the main urban settlement, and the town of Rio in the northern part of the graph.

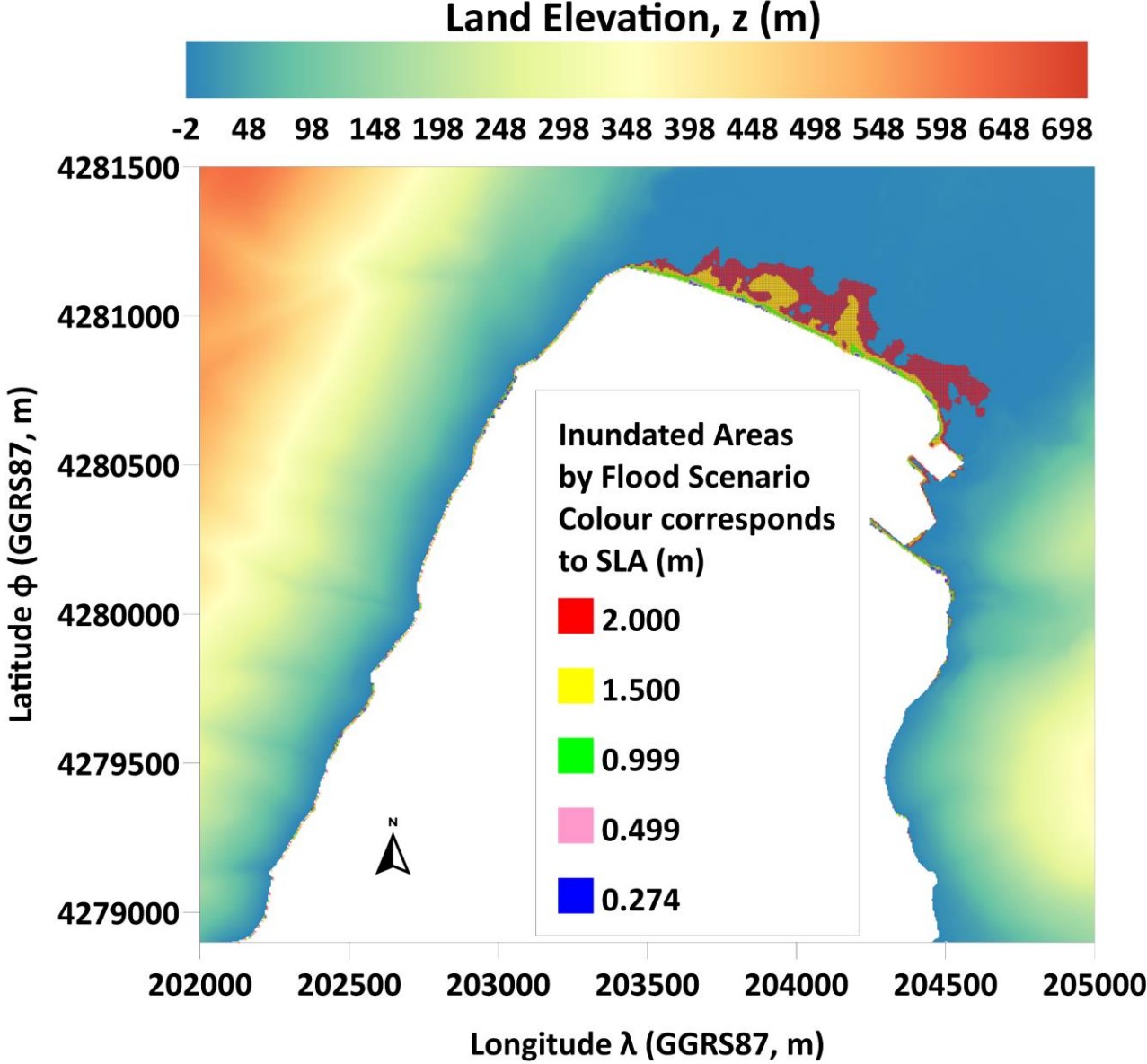

**Figure 15.** Map of estimated flooded areas as depicted by operational CoastFLOOD simulations, driven by an in situ recorded SLA = 0.274 m and four extreme case scenarios of TWL = 0.5–2.0 m, for the coastal study area of Vassiliki bay (Area 2; south-western Lefkada Island), also including a small fishing harbour port in the north-eastern part of the bay.

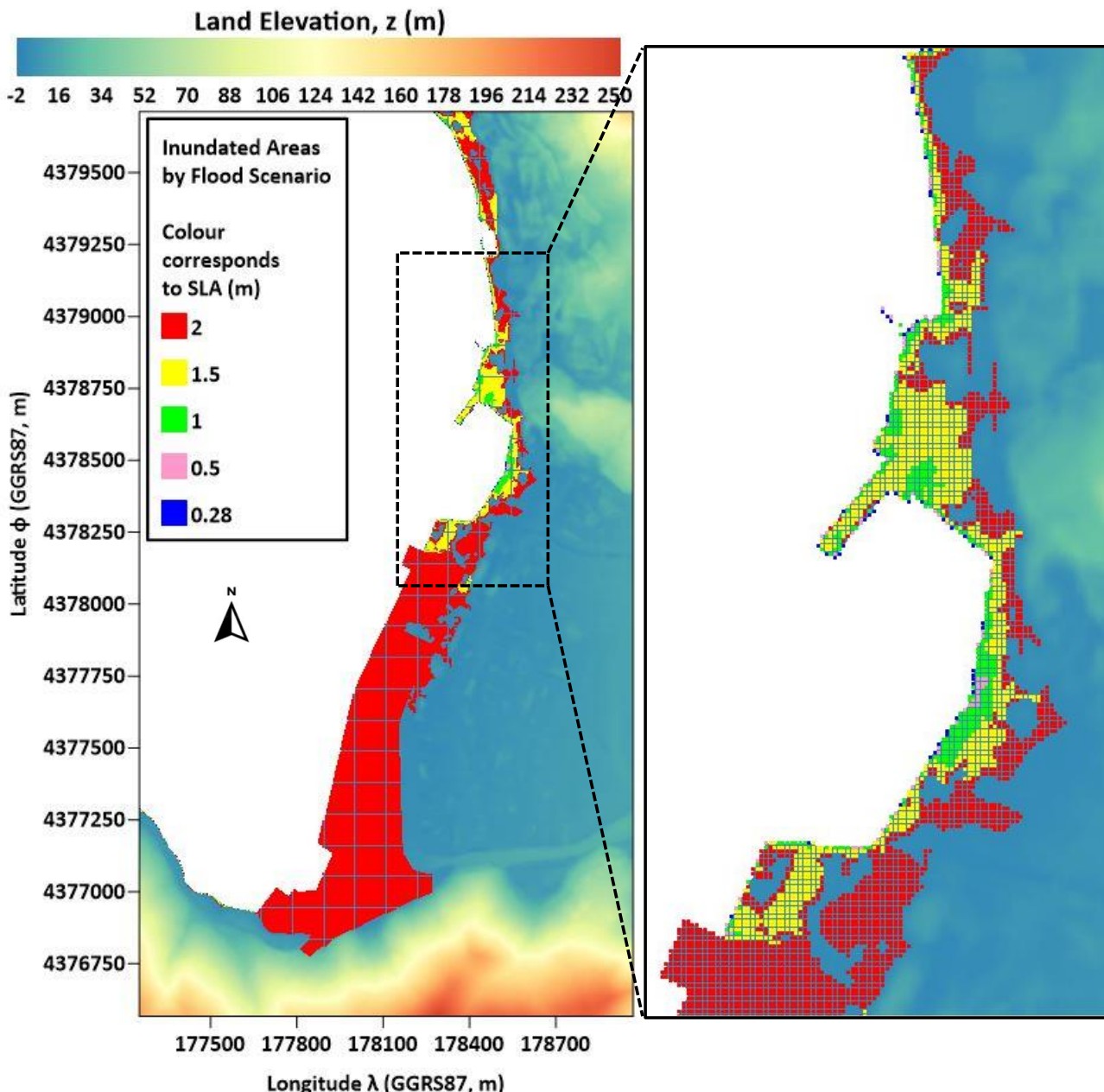

**Figure 16.** Map of estimated flooded areas as depicted by operational CoastFLOOD simulations, driven by an in situ recorded SLA = 0.28 m and four extreme case scenarios of TWL = 0.5–2.0 m, for the coastal town of Igoumenitsa port (Area 4; North-western Epirus) with its port. Insert map presents the zoomed-in depiction of the northern port area.

An interesting feature is the formulation of the timeframe for maximum flood inundation reach, $t_{MIR}$, in some study cases. The pattern of $t_{MIR}$ is similar and, in general, increasing for the ascending values of $SLA_{max}$ = 0.2–2 m, except from the Laganas and Kyparissia case studies (highlighted in Table 4) and the lower values of recorded $SLA_{max}$ = 0.2–0.3 m, for which $t_{MIR}$ is counterintuitively quite high; i.e., larger than the $t_{MIR}$ of larger SLAs and consequent inundation extents. However, this is probably reasonable because lower SLA values on the coastline drive much slower inundation flows than larger storm surge levels, since shoreline SLA/SSH acts as the main formulation factor of the hydraulic head of the flood front propagation. The latter is valid given the peculiarities of the topographic formulation of the studied area. Nevertheless, this fact reveals that the CoastFLOOD model, with proper treatment of the bottom roughness (Manning coefficient

*n*), can produce rather plausible estimations of the time evolution of flood inundation phenomena.

## 5. Discussion

During the last two decades, with the rise in available computational power and resources, the approach of reduced complexity in flood modelling has become the norm for the estimation of coastal inundation due to sea level increase and the lack of available field observations, in order to support Integrated Coastal Zone Management (ICZM) and strategic decision-making. ICZM requires spatiotemporally broad estimations in large-scale domains, $O(x) = 10–100$ km, yet with high-resolution modeling on grid cells with $O(dx) = 1$ m. High-frequency and/or robust updated field data (topographies, transient water areas, reliable DEM/DSM, etc.) in the highly changing coastal zone are hitherto rather limited, making it difficult to feasibly apply the multiple fine-resolution simulations needed at a regional scale in littoral areas. To this end, proper hydraulic models with quick solvers, that neglect secondary effects of turbulence, hypercritical flows, local acceleration terms in the momentum equations, water infiltration and percolation at the bed, etc., such as CoastFLOOD, presented herein, can provide a computationally viable alternative for modelling flood inundation in the coastal environment [26,83,94,99].

The x- and y-direction decoupling of the lower order semi-analytical flow equations in such models may undermine the reproduction of diffusive effects in the hydrodynamic flow of floodwater masses, but the proposed approach is rather simple and allows for easy numerical coding that is computationally robust and produces very similar results to more sophisticated models for flood wave propagation [58,60,83,100]. Thus, on each grid element, the mass and momentum conservation principles are translated into simplified semi-analytic hydraulic equations for continuity (based on floodwater depth and hydraulic head calculation) and volumetric flow rates (Manning-type flow driven by a hydrostatic approach for the piezometric load and bottom friction). These can be separately solved on the centre and faces of the grid cells of a finely discretized domain. The main advantage of such a method for flood routing is the easy use of a wet/dry cell storage module [58,91].

The main disadvantages of reduced complexity flood models are the oversight of sub-grid scale features of the flow (e.g., cavitation, recirculation, aeration, debris advection, and viscosity effects) [144] and fine-scale spatial features (e.g., drainage systems, sewers, conduits, bridge culverts, pools, and drillings). Nevertheless, if one needs to find spatially broad-scale information regarding the inundated areas' extents and the floodwater level in them, and not the full details of the transient flood hydrodynamics, then neglecting the aforementioned effects on the flow is plausible. The secondary fine-scale topographical features of small engineering structures (open canals and conduits, etc.) should play a role in properly modelling the flood flow only in the beginning of the inundation process, when these technical structures are empty and have adequate depth. After enough time, these open channel formations become filled either with rainwater or with seawater, allowing the floodwater to only flow above the hydraulic structures' crests, and this is what we approach herein. Another relevant issue is the exclusion of floodwater percolative interaction with the porous bed and the downward infiltration to the aquifer. However, these flows are usually very slow processes compared to the hydraulic propagation of flood fronts, and thus they cannot significantly influence the hydrodynamics of inundation (this might not be the case for extreme TWL > 1 m in Patra city, where floods that reach maximum duration might range between 2–3.5 days; Table 4). Moreover, the soil on which the floodwater propagates should probably be saturated with rainwater from the storm. Hence, seawater should flow as a runoff on the floodplain's saturated ground surface. Furthermore, inundation in coastal areas is apparently a combined result of river/watershed, precipitation, and ocean (compound) flooding. Therefore, there is a need to integrate fluvial floods with (pluvial) surface runoff and coastal water run-up in order to model flood inundation in littoral lowlands.

A matter that may cause uncertainties in coastal flood flow prediction in urban environments is the depiction of topographic details that are finer than the available DEM/DSM resolution or their vertical accuracy; i.e., the inclusion of outdoor microstructures (uneven pavements, sidewalk fringes, raised curbs, fences, roadblocks, bumps, and obstacles, etc.), stairs, gates, doors, and basement windows at ground level. These can either prevent the free flood flow or absorb floodwater, draining it inside buildings and basements. These effects cannot be taken into account by the model but seem to only be significant in densely built/populated urban spaces and not on coastal floodplains, such as the studied coastal zones of the Ionian Sea that were presented here (excluding Patra city; Area 10). The typical model grid cell should not exceed the upper thresholds of O($dx$) = 10–100 m given that the characteristic flood flow depths range between 0.1–1.5 m and Manning's $n$ fluctuates between 0.001–0.4 s·m$^{-1/3}$, respectively. The larger spatial discretization step used herein is $dx$ = 5 m, which is considered quite fine. Consequently, the choice of a zero-inertia model that can reduce the complexity of floodplain hydraulics to an imperative minimum representation of the flow equations is acceptable for slow (big volumetric flow changes occur in timescales >> $dt$) and shallow (vertical changes in floodwater flow depths are practically a lot smaller than horizontal ones or the typical cell width $dh << dx$) flood flows [58]. Neglecting inertia terms can only play a local role, in the sense that the ability of 2-D reduced complexity models to reproduce flood propagation has been corroborated by several researchers in the past based on comparisons with available field data and other model approaches [71,83,90]. It is clear that the spatial resolution and the consequent timestep of the numerical solution are the most crucial factors in defining robust simulations for this kind of reductionistic modelling approaches. These issues are adequately addressed in the CoastFLOOD simulator, offering computational efficiency, ease of coding for GIS raster-based applications, broad-scale (regional flood reach) simulations, and repeatability from a pragmatic management perspective for engineers, scientists, and managing authorities.

The lack of field data for calibration and validation may be the major constraint in the further verification of reduced complexity flood inundation models for coastal areas. The recent evolution of remote sensing products and their available resolutions seems to partially address this issue in a qualitative manner. The inherent discrepancies to distinguish the source of floodwater (e.g., tidal surge, wave action, drainage or runoff, and rainfall) is a problem for the quantitative validation of coastal flooding modelling due to storm tides in tandem with MSLR [71,145]. Therefore, we also compared our hydraulic flood model results with a Bathtub-HC approach. However, when using the latter, one should consider issues arising from the omission of bottom friction leading the analysis by exaggerated flood vulnerability estimations. Several coastal managers have inferred that the latter can lead to overprotective engineering solutions, excessive defence schemes, and inflated investment against flood protection. Despite this, we believe that a Bathtub-HC method should always be applied to indicate low-lying flood-prone areas in the coastal zone to formulate an idea about potentially inundated areas and to direct the more focused (high-resolution) coastal flooding approaches under extreme sea level elevation in the future.

Finally, model implementations in areas that are too large might require rather large timestep values (given the available computational resources and timeframes, especially in operational mode), which may lead to chequerboard-type oscillations in the numerical solution, not easily suppressed or relaxed, especially in areas with small gradients of the floodwater free-surface and subsequent slow evolution of the flow. CoastFLOOD solves this issue with the use of a proper CFL criterion within an adaptive time-stepping algorithm [27,33,71,90,91].

Thus, CoastFLOOD has been recently upgraded to include very detailed depictions of bottom roughness (based on recently available land cover data), the influence of storm surge-led currents on the coastline boundary, fine-scale DEM/DSM, and the enhancement of wet/dry cell techniques for flood front propagation over steep water slopes. These

techniques have been proposed by other researchers in the past, and we included them as options in this new updated code. An additional novelty is related to the very fine-scale DEM/DSM of $dx$ = 2 m, providing high detail of the domain terrain. Moreover, a cross-type scan of the model grid (N→S/S→N in the meridional direction; W→E/E→W in the zonal direction) is now applied in every timestep, thus allowing for plausible estimations of the flood front propagation from any direction of the horizon or peripheral boundary, while some coastal inundation models still only allow one-way flood propagation; i.e., either from south/north or west/east.

Future research should include even finer scale simulations and comparisons with model formulations considering local acceleration terms in tandem with the proper depiction of details over and around coastal structures, port infrastructure, beach land formations, and rocky shores in the model grid. The treatment of sub-grid topographical features (weirs, drainage holes on embankments, drainage trenches and channels under bridges, sewerage networks, etc.) should also be included in future developments of the CoastFLOOD model. Incorporating a breaching mechanism for sand dunes and coastal embankments should also be implemented. An additional consideration is to combine a percolation and subterranean infiltration module to account for ground porosity effects on the floodplain, together with a simple approach for the evaporation of inundated seawater. The latter can always contribute to more long-term simulations that may result in different patterns of floodplain water storage, floodwater encroachment and conveyance, as well as possible backwater effects from flood flux blockage, etc.

Therefore, we believe that, although it presents no ground-breaking scientific novelty, it provides very much needed technical innovations, i.e., a first national-level OFP for surge-induced coastal floods established in Greece since the 1980's concepts of flood hydraulics for coastal (optionally combined with fluvial-deltaic) inundation by storm surges and sea level elevation in general.

## 6. Conclusions

In this study, we present applications using a new code (CoastFLOOD), developed in FORTRAN-95, for a classic modelling approach of 2-D hydraulic flood flow in coastal areas. CoastFLOOD is built on the concept of high-resolution, storage-cell, mass balance flood inundation for coastal lowlands, following the simplified approach of Manning-type flow equation, under a reduced complexity concept, running on a GIS raster-based domain. Although a detailed physical representation of turbulent floodwater hydrodynamics is overlooked, CoastFLOOD relies on computational efficiency and the delivery of stable simulations with robust results. The model's performance is evaluated for the case of predicted (i.e., ocean modelling) or observed (i.e., satellite altimetry) storm surges affected by tidal components of sea level elevation (also termed as storm tides). The proposed methodology and numerical model could be applied in operational applications as well as studies of long-term mean sea level rise or short-term extreme scenarios of total water levels, also considering an estimative mean condition for wave runup, but mainly excluding the high-frequency phenomena, such as the undulating sea surface uprush and backwater effects due to waves, etc.

The flood extent identification was based on the computation of the NDWI index derived from remote sensing ocean color data by Sentinel-2 satellite. The verification of the model was performed for two cases of recorded storm surges in the Ionian Sea; the first during a storm in December 2021 in the Manolada-Lechaina coastal zone (Area 1; north-western Peloponnese, western Greece), and the second in September 2020 during the Ianos Medicane landfall in Livadi bay (Area 5; southern Cephalonia Island; Androulidakis et al., 2023). The comparison of CoastFLOOD simulation results against NDWI-identified flooded areas show that our model can reproduce the coastal flooding mechanism in areas that are more-or-less affected (wetted) by stormy weather during the timeframe of analysis. The model results maybe overpredict the recorded flood extents because the satellite data are not totally accurate to represent the actual situation of

floodwater extents during the storm surge, since the satellite does not usually coincide with the peak of the storm surge due to cloud contamination, and thus, it is not representative of the maximum flood reach. In the model's defence, the predicted flood extents on the southern coastal zone of Cephalonia Island (Area 5) definitely overlap and include the wet areas traced by remote sensing, and that is on the safe side in terms of engineering and coastal management. Moreover, some available soft data (visual proof and pictures from social and mass media reports) can also be used to corroborate the general performance of the model [30].

The validation of the CoastFLOOD model's efficiency to reproduce the highest possible flood extent in coastal plains was also tested against an efficient Bathtub-HC approach. The agreement between the two approaches is quite high with very high *GoF* [30,91,94] scores (>0.95) for both the realistic sea level and extreme scenario TWL cases. Furthermore, we show that proper treatment of the bottom roughness with spatially distributed Manning coefficients referring to realistic land cover datasets can formulate a more realistic estimation of the timeframe for reaching maximum flood inundation extents. Therefore, the bottom friction parameter is defined as the main calibration feature. The realistic reproduction of the flooded inland areas' roughness, based on different representations of the land cover information by CLC datasets, was investigated in detail. Specifically, we created a matching list of all CLC-2018 codes to a detailed set of discrete types for earth/ground material that correspond to a detailed list of different assigned Manning coefficient values in the CoastFLOOD model. The use of a horizontally distributed field of gridded Manning coefficient values (based on the CLC) compared to a global effective value of a grid-scale Manning coefficient did not highly affect the estimation of the flood extent and the location of impacted areas in agreement with previous studies [94]. However, it drastically influenced the calculation of the timespan for maximum flood reach. Moreover, it was shown that the latter heavily depends on the levels of the storm-induced sea level on the coastline, which acts as the hydraulic head of flood front propagation; i.e., lower storm surge heights may drive much slower inundation flows than larger ones. Hence, the proposed model also shows an intuitively correct sensitivity to realistic representations of floodplain friction, especially if it is applied in areas with complex topographies. The use of highly variable friction coefficients for coastal flood modelling should provide better predictions for the duration of an inundation event, which is crucial to first-level responders and coastal zone managers. Still, it is concluded that the detailed depiction of topography is the key constraint on robustly formulating and realistically simulating the floodwater flow for the accurate determination of the maximum flood extent.

The most probable explanation for any discrepancy in comparisons of modelled and observed flood extents in the coastal zone is the uncertainty of field data concerning the actually occurred flood rates. Thus, large uncertainties of the latter, mainly stemming from the sources of seawater inundation, except from storm tides, e.g., wind waves and swell, make it difficult to develop a definite benchmark case dataset with which to robustly test the performance of storm-induced coastal inundation models. Indeed, it has been argued [30] that for random coastal inundation events, storm surge flooding usually coincides with wave overtopping, making it very difficult to produce any reliable observation dataset capable of being used as a reference against competing coastal model formulations in a meaningful way. Hence, as a future research step, there is a need to incorporate a treatment of boundary conditions in the CoastFLOOD model as a varying timeseries of non-deterministic values in order to avoid substantial underestimations of coastal inundation and potentially relevant risk.

The Ionian Sea's coastal zone in Greece is eventually threatened by storm surge inundation in an annual cycle, with likely coastal flooding events occurring during mid-autumn (late September—early- to mid-October) and during December or early January, as also pointed out in [17,23,118–120]. The impacts are not very pronounced for usual storm surge levels (<0.3 m) but can be severe for extreme cases of total water levels (>0.6

m as found in future climatic projections along the Greek coastal zone; Makris et al., 2016; Galiatsatou et al., 2019, 2021), e.g., in coastal urban areas (Igoumenitsa, Patra, and Kalamata).

Conclusively, we presented a robust, easy-to-use, numerical tool for coastal inundation due to storm surge/tide flooding, under the reduced complexity notion, imperatively needed for operational forecasts of storm impact. Nonetheless, it can hopefully be useful for both operational applications and projected climatic studies of coastal inundation under extreme scenarios to help coastal zone managers, policymakers, and involved stakeholders to better estimate the characteristics of coastal (or compound) flooding under conditions of environmental change.

**Author Contributions:** C.M.: Conceptualization, Formal analysis, Investigation, Methodology, Software, Validation, Visualization, Writing—original draft, Writing—review and editing; Z.M.: Data curation, Methodology, Software, Validation, Visualization, Writing—review and editing; Y.A.: Conceptualization, Data curation, Formal analysis Investigation, Methodology, Software, Writing—review and editing; Y.K.: Conceptualization, Investigation, Project administration, Resources, Supervision, Writing—review and editing. All authors have read and agreed to the published version of the manuscript.

**Funding:** This research received no external funding.

**Data Availability Statement:** The data presented in this study are available on request from the corresponding author. The data are not publicly available due to copyright restrictions of intellectual property produced within AUTh.

**Conflicts of Interest:** The authors declare no conflict of interest.

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
