# Peer review of "CoastFLOOD: A High-Resolution Model for the Simulation of Coastal Inundation Due to Storm Surges"

_hydrology, doi:10.3390/hydrology10050103_

Round 1

Reviewer 1 Report

GENERAL COMMENTS:

The manuscript presents a raster-based inundation model applied to coastal flooding.
In general, the paper is well organized but it is too long and repetitive: for instance, the “Introduction” covers 3 pages and the “Conclusions” covers 2 pages plus 2 pages of “Discussion”. Also, the abstract is too long: the abstract should be an objective representation of the article. I think that the manuscript requires a first large revision to summarize some concepts and delete superfluous or repeated sentences.

In addition, I don’t understand why the authors choose the journal “Hydrology” instead of “Water” from the same publisher that I think fit better the topic of the manuscript.

The model proposed by the authors, which solves a “simplified form for the Shallow Water Equations”, is very similar to others already published. As mentioned by the authors, the model is similar (maybe the same) to the LISFLOOD one proposed by Bates and De Roo, 2000. In fact, it is not clear the novelties and peculiarities of the COASTFLOOD model with respect to the LISFLOOD model.

Moreover, other authors proposed raster models that are similar to the LISFLOOD one with some modifications, that should be cited. For instance, the model proposed by the research group of Favaretto, C., Martinelli, L., & Ruol, P. (2019). A model of coastal flooding using linearized bottom friction and its application to a case study in Caorle, Venice, Italy. International Journal of Offshore and Polar Engineering, 29(02), 182-190.

Therefore, the authors are encouraged to highlight more clearly the specific novelties and differences of the model with respect to the other cited models.

Finally, it is not clear if the model has been tested with some benchmarks to validate it. For instance, some well-known benchmarks (analytical or experiments) for SWE equations in the coastal flooding context are the free-surface sloshing with friction or the solitary wave propagation around a circular island (Briggs et al., 1995).

For these reasons, I suggest a major review of the paper.

SPECIFIC COMMENTS:

[a] Line 49: “physical and technical barriers” What do the authors mean by these terms?

[b] Line 57-60: Several references (11) are cited relative to the general issue of coastal flooding in low-lying littoral areas. However, this way of quoting does not highlight the specific contributions and peculiarities of the different articles and does not recognize the work done by several authors.

[c] The quality of the figures is very poor. The legends are unclear and the choice of colors is ineffective. The authors said in the figures caption that “good agreements” are visible but the comparisons among the model results, the NDWI and the bathtub approach are not possible with this poor quality.

Author Response

Attached please find the answers to all reviewers' comments.
Thank you very much for reviewing our submitted manuscript. 

Reviewer 2 Report

In work entitled " CoastFLOOD: a high-resolution model for the simulation of coastal inundation due to storm surges" by XYZ et al., the Authors proposed numerical code (CoastFLOOD) performs 10 high-resolution modelling of coastal floods induced by storm surges which are very interesting need of the demand.  I have read the manuscript, and I saw that its results fall within the scope of the journal.  The authors have well written the manuscript and the figures are of good quality. However, need some minor modification and reformulation of sentences before its eventual acceptance.

Abstract:

Comment:1 Overall, the abstract is well written and summarizes the findings of the studies. However, has some scope for improvement to improve readability and also include more findings replacing the very general discussion that is actually part of the methodology.

Comment 2: GIS should not be the keyword.

Comment:3 Introduction needs to add recent references and includes some discussion of the motivation of the study at the end of the introduction.

Comment: 4 The precipitation extremes, particularly with associated storms or cyclone play, contribute significantly to the storm surge. The Strom density shows strong spatial variability over Mediterranean regions, and due to climate change, the storms are expected to alter significantly,  which can alter the storm surges induced CoastFLOOD (Lionello et al. 2020; Mishra et al., 2023; Reale et al., 2022). Authors can include this discussion.

Mishra, A. K.Jangir, B., & Strobach, E. (2023). Does increasing climate model horizontal resolution be beneficial for the Mediterranean region?: Multimodel evaluation framework for High-Resolution Model Intercomparison ProjectJournal of Geophysical Research: Atmospheres128, e2022JD037812. https://doi.org/10.1029/2022JD037812

Reale, M., Cabos Narvaez, W.D., Cavicchia, L. et al. Future projections of Mediterranean cyclone characteristics using the Med-CORDEX ensemble of coupled regional climate system models. Clim Dyn 58, 2501–2524 (2022). https://doi.org/10.1007/s00382-021-06018-x

Lionello P, Barriopedro D, Ferrarin D, Nicholls CRJ, Orlic M, Raicich F, Reale M, Umgiesser G, Vousdoukas M, Zanchettin D (2020) Extremes floods of Venice: characteristics, dynamics, past and future evolution. Nat Hazards Earth Syst Sci Discuss. https://doi.org/10.5194/nhess-2020-359 

Comment:5 Author can tabulate the previous literature (Line 104-124 ) along with corresponding details.

Comment:6 Typo: Dscription > Description

Comment:7 Some of the details can be provided as supplementary: for example:

(i)             Description of areas’ characteristics

(ii)           Description of CLC label areas’ characteristics

Comment:8 Conclusion is too short. I would suggest adding more information.

Author Response

Attached please find the answers to all reviewers' comments.

Thank you for reviewing our submitted manuscript.

Round 2

Reviewer 1 Report

Thank you for your edits and replies in response to my comments.
It is my opinion that the revised manuscript is much improved in terms of readability and clarity and I now find it suitable for publication.